# Future Dynamic 3D Reconstruction:
# A 3D World Model with Disentangled Ego-Motion

**Nils Morbitzer** [* 1 2]   **Jonathan Evers** [* 1 3]   **Artem Savkin** [3]   **Thomas Stauner** [3]
**Nassir Navab** [1 2]   **Federico Tombari** [1 2]   **Stefano Gasperini** [1 2 4]

## Abstract

Forecasting the evolution of dynamic environments is crucial for autonomous agents. While generative world models have recently achieved high photorealism in 2D video synthesis by mixing ego-motion and environmental dynamics within the image plane, they exhibit physical inconsistencies, such as morphing or vanishing objects, especially over long time horizons. In this paper, we propose FR3D, a world model that predicts a persistent 3D latent representation for future dynamic 3D reconstruction. Unlike prior works that treat the world as a sequence of image-based features, FR3D explicitly decouples the 3D evolution of the scene from the agent's trajectory, treating the inferred ego-motion as a latent proxy for action. This disentanglement resolves the ambiguities between self-motion and world-motion, ensuring geometric consistency into the future. Furthermore, we introduce a teacher-student distillation strategy that leverages the spatial "common sense" of off-the-shelf foundation models, leading to robust zero-shot generalization. Extensive experiments demonstrate FR3D's strong performance for future dynamic 3D reconstruction from monocular observations across multiple datasets, even 2 seconds into the future. Project page: https://fr3d-wm.github.io.

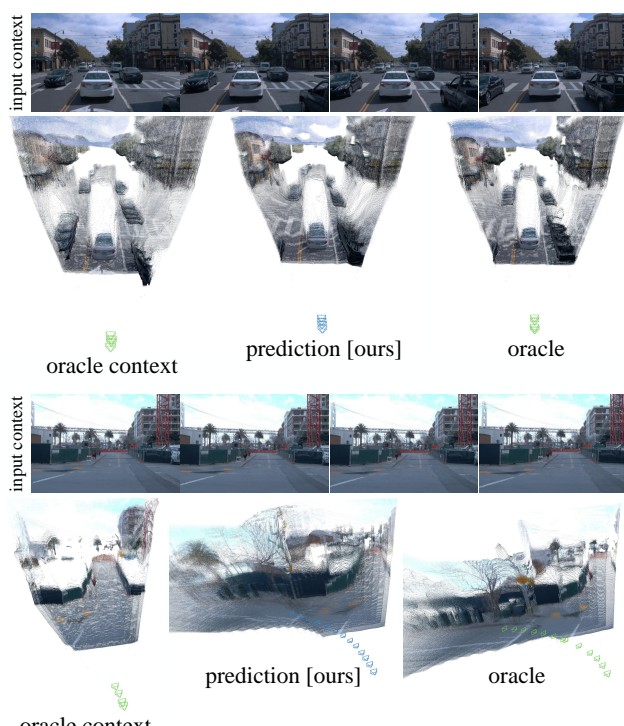

*Figure 1.* The proposed FR3D is a 3D world model predicting future 3D reconstruction of dynamic scenes that takes monocular images as input. FR3D disentangles the forecasting of the induced ego-camera motion from that of the 3D scene structure. As shown in these future predictions of challenging scenes, FR3D successfully handles dynamic scenes with traffic in both directions (above) and estimates turning events smoothly (below).

## 1. Introduction

Anticipating future events is a key ability for intelligent agents operating in complex environments, from robotics to autonomous navigation (Dosovitskiy & Koltun, 2017; Hu et al., 2020; 2021). To estimate the unfolding of these events, agents rely on a world model: an internal engine designed to capture the underlying dynamics of the environment. Ha and Schmidhuber (2018) originally formalized the concept of a world model as a generative simulator that directly synthesizes the evolution of the environment in sensor space. Conversely, more recent perspectives by LeCun (2022; 2024) argue that generative photorealism is often irrelevant, advocating instead for world models that

---
[*]Equal contribution   [1]Technical University of Munich (TUM) [2]Munich Center for Machine Learning (MCML) [3]BMW Group [4]Visualais.   Correspondence to: Nils Morbitzer <nils.morbitzer@tum.de>.

*Proceedings of the 43rd International Conference on Machine Learning*, Seoul, South Korea. PMLR 306, 2026. Copyright 2026 by the author(s).

prioritize abstract, task-relevant predictive representations in latent space. Despite these divergent philosophies on *how* to represent the world, a key consensus remains: a world model's core capability is the accurate estimation of future states given a history of observations and potential actions.

In recent years, generative world models have demonstrated a remarkable success at simulating complex environments, with 2D video diffusion models achieving unprecedented levels of photorealism (Brooks et al., 2024; Agarwal et al., 2025; Google DeepMind, 2025). These 2D generative models are particularly useful as interactive environments for video games (Bruce et al., 2024) or controlled experimental environments in real-world simulators for autonomous systems (Hu et al., 2023). However, by operating strictly in the image plane, these models mix camera trajectories with scene evolution. This entanglement fundamentally limits the model's ability to maintain a coherent 3D geometric structure throughout the rollout, often leading to "hallucinated" physics where objects morph or vanish due to ego-motion (Hu et al., 2023).

For an agent to interact meaningfully with the physical world, e.g., performing tasks in it, such as in autonomous driving, geometric integrity must take precedence over photorealism. Without a strong 3D inductive bias, temporal rollouts inevitably lead to inconsistent depth-dependent motion parallax and violations of object consistency. While most current world models operate only in pixel space or image-based feature space (Hafner et al., 2023; Bardes et al., 2024; Wang et al., 2024b; Zhou et al., 2025; Baldassarre et al., 2025), treating the world as a sequence of 2D planes, the physical world is inherently 3-dimensional, leading to inconsistencies, especially over longer time horizons (Xiao et al., 2024). Instead, attempts in 3D have mostly focused on occupancy prediction (Zheng et al., 2024), on the synthesis of sensor data (Zhang et al., 2024), or do not disentangle ego-motion and world-motion (Hu et al., 2021). Furthermore, existing state-of-the-art models achieve generalization only through extreme scale, requiring even 22M GPU hours and 20M hours of video data (Agarwal et al., 2025), making them highly impractical for specific downstream tasks.

In this work, we bridge the gap between dynamic 3D reconstruction and latent world modeling. While 3D reconstruction is retrospective and prior world models focus on 2D pixels or mix ego-motion with world-motion, we introduce a model that maintains a persistent 3D latent representation into the future. As shown in Figure 1, our model explicitly decouples the evolution of the 3D scene from the agent's future trajectory within the same coordinate frame, treating the inferred ego-motion as a latent proxy for action. By disentangling ego-motion from world-motion, the model resolves the fundamental ambiguity between the changes caused by the self (ego-motion) and those caused by external entities

(world-motion). Additionally, to bypass the prohibitive costs of large-scale training, we propose a teacher-student distillation strategy that leverages the spatial "common sense" of off-the-shelf foundation models. This allows our model to achieve robust zero-shot generalization with significantly lower data and compute requirements. We term our method FR3D, for future dynamic 3D reconstruction, and we summarize our contributions as follows:

- We introduce the task of future dynamic 3D reconstruction from monocular observations, leading to a new paradigm for 3D world models.

- We present FR3D, the first world model for dynamic 3D reconstruction that learns to propagate scene states and agent poses within a unified 3D latent space.

- We show that disentangling ego-motion from environmental dynamics enables significantly more stable long-horizon predictions, even 2 seconds ahead.

- We introduce an efficient training strategy that achieves strong zero-shot performance by distilling geometric priors from foundation models.

## 2. Related Work

### 2.1. Feature Prediction World Models

An emerging paradigm for future-frame prediction and scene understanding focuses on estimating future visual features instead of raw RGB values (Karypidis et al., 2025a; Zhou et al., 2025; Baldassarre et al., 2025; Boduljak et al., 2025). FUTURIST (Karypidis et al., 2025b) introduces a multi-modal forecasting strategy for depth and semantic segmentation. Despite impressive results, it relies on task- and dataset-specific encoders, limiting its generalization. DINO-Foresight (Karypidis et al., 2025a), DINO-WM (Zhou et al., 2025), DINO-world (Baldassarre et al., 2025) demonstrate that pre-trained, general-purpose features of established visual foundation models (VFM) can serve as a state space for world models. While the former focuses on common scene understanding tasks such as semantic segmentation, depth, and surface normals, the latter introduces an approach for modeling visual dynamics via 2D RGB reconstruction. While previously mentioned approaches perform deterministic forecasting, VFMF (Boduljak et al., 2025) proposes a generative forecaster to model the probabilistic nature of future predictions. To achieve this, the authors encode VFM features into a compact latent space suitable for diffusion.

Whereas prior works (Karypidis et al., 2025b;a; Boduljak et al., 2025) mainly focus on forecasting 2D or 2.5D tasks, we address the problem of future 3D scene prediction and reconstruction by forecasting both ego-camera motion and 3D scene dynamics, yielding strong forecasting performance

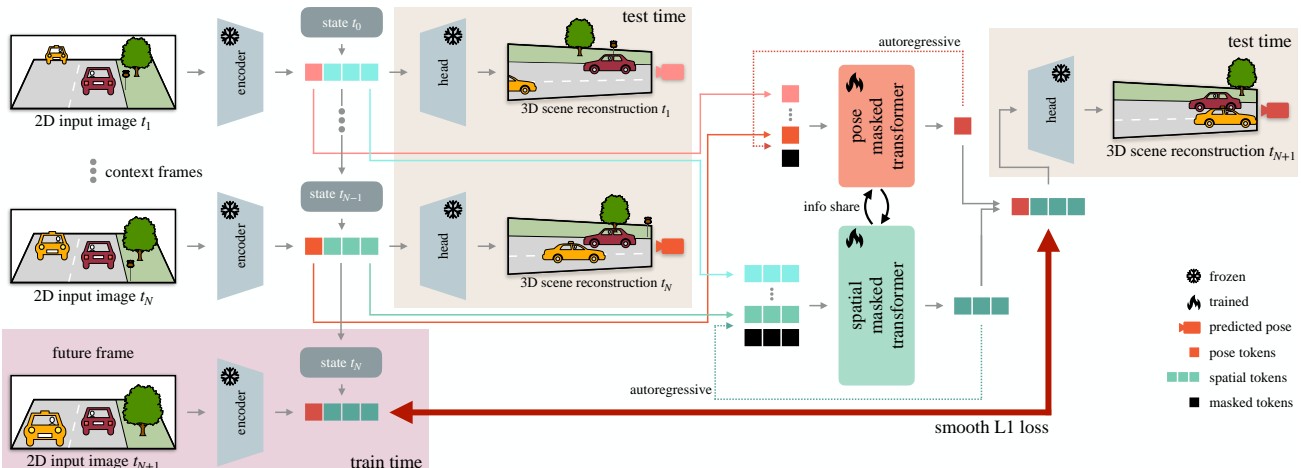

*Figure 2.* The proposed FR3D takes in input a sequence of images as context (up to time $t_N$), and outputs a unified 3D scene reconstruction with ego camera poses autoregressively for the next timestamps (from $t_{N+1}$ onwards) without accessing the corresponding images. Tokens and state are internal representations of the scene from previous frames, and the model estimates future tokens and decodes them into a 3D reconstruction by leveraging an off-the-shelf foundation model (its encoder, its decoder used to combine the current tokens with the state, and its heads), such as CUT3R (Wang et al., 2025b). The future prediction is performed by two masked transformers: one for ego poses and one for scene geometry. FR3D is trained autoregressively in a teacher-student paradigm by mimicking the token space of the frozen foundation model via a smooth L1 loss.

even when reasoning 2 seconds into the future. In addition, prior work still relies on dataset-specific decoders/heads to make its predictions interpretable. Instead, we aim for zero-shot generalizable performance. To address this, we propose injecting our approach to estimate all the scene information necessary for 3D reconstruction in a pre-trained reconstruction model's latent space and use its decoders/heads to make our latent predictions interpretable.

## 2.2. 3D World Models

World modeling has shifted from 2D pixel synthesis to spatially aware 3D representations, improving geometric consistency. For example, Copilot4D (Zhang et al., 2024) formulates future scene prediction as discrete diffusion over point clouds, whereas DiST-4D (Guo et al., 2025) adopts a disentangled spatiotemporal diffusion design that incorporates LiDAR-derived metric depth for 4D scene generation. In contrast, OccWorld (Zheng et al., 2024) predicts future 4D occupancy grids to densely model how scenes evolve. Drive-OccWorld (Yang et al., 2025) integrates action conditioning to enable controllable generation and leverages this generative capability for end-to-end planning. While these models show a significant leap forward, they often rely on expensive occupancy annotations or sensor data such as LiDAR point clouds, or fail to explicitly disentangle the ego-motion from the environment's dynamics. Instead, FR3D introduces a unified 3D latent space that decouples these two motion components, enabling more stable long-horizon predictions.

## 2.3. Feed-Forward 3D Reconstruction

Traditional geometry-based pipelines, such as Structure-from-Motion (Schönberger & Frahm, 2016) and Multi-View Stereo (Schönberger et al., 2016), tackle 3D reconstruction through a sequence of interdependent subproblems, including feature matching, pose estimation, and depth estimation. Because each stage is not solved perfectly and depends on the estimated outputs of preceding steps, errors inevitably accumulate and propagate, often degrading the quality of the final reconstruction.

More recently, learning-based approaches have overcome these limitations by jointly addressing multiple reconstruction tasks. DUSt3R (Wang et al., 2024a) introduces a transformer-based framework that directly predicts dense 3D pointmaps from image pairs, effectively associating 2D pixels with 3D geometry in a feed-forward manner. This representation enables the estimation of complete scene parameters, including 3D structure, camera poses, and intrinsics. Building on this, MASt3R (Leroy et al., 2024) improves DUSt3R's matching accuracy by incorporating a local dense feature head and explicitly training for metric 3D reconstruction. VGGT (Wang et al., 2025a) demonstrates that scenes can be reconstructed from an arbitrary number of images without additional global optimization.

While DUSt3R (Wang et al., 2024a) and many subsequent works focus on offline 3D reconstruction, where all observations are available a priori, online 3D reconstruction methods aim to incrementally recover scene geometry from a continuous stream of incoming data. To this end,

Spann3R (Wang & Agapito, 2025) introduces a spatial memory mechanism that learns to retain and update information from past observations, enabling progressive scene reconstruction. CUT3R (Wang et al., 2025b) proposes a compact latent state representation that not only stores previously observed geometry but can also be read out to reason about unobserved regions given a camera pose.

While previous feed-forward 3D reconstruction methods focus on reconstructing already observed information, our approach forecasts future 3D scene information, including static and dynamic 3D structures and the camera poses.

## 3. Method

### 3.1. Preliminaries

In a nutshell, a world model predicts how an environment evolves over time from observations, actions, and uncertainty. Let $t$ denote a discrete time step, $x_t$ the observation at time $t$, $a_t$ the action taken, and $s_t$ the underlying system state. The observation is first encoded as

$$h_t = \text{Encoder}(x_t).$$

Given the encoded observation $h_t$, the current state $s_t$, the action $a_t$, and a latent variable $u_t$, a predictor models the state transition as

$$s_{t+1} = \text{Predictor}(h_t, s_t, a_t, u_t).$$

The latent variable $u_t$ represents unobserved or unknown factors, enabling the model to capture a distribution over plausible future states rather than a single deterministic outcome (LeCun, 2022).

### 3.2. Future 3D Scene Prediction

Our approach targets predicting how a 3D scene evolves over time and forecasting the most likely camera motion. Displayed in Figure 2, we propose a world modeling approach for dynamic 3D reconstruction. Our key idea is to learn how to temporally operate in the latent space of a pre-trained, feed-forward 3D reconstruction model, thereby effectively disentangling camera motion from the reconstructed 3D scene structure and potentially induced motion. For that, we propose an autoregressive teacher-student distillation training strategy that yields the first 3D world model for dynamic 3D reconstruction learning to propagate scene states and agent poses within a unified 3D latent space.

Let $I \in \mathbb{R}^{N \times 3 \times H \times W}$ denote a sequence of $N$ context images $I_t$ of size $H \times W$ where $t \in [1, N]$ represents time. As depicted in Figure 2, we first encode $I$ (observations) using a pre-defined encoder $\mathcal{E}$ of a pre-trained, feed-forward 3D reconstruction model $R$ yielding per-frame image tokens:

$$F = \mathcal{E}(I) \in \mathbb{R}^{N \times D \times H_F \times W_F} \tag{1}$$

where $D$ is the image token dimension and $H_F \times W_F$ are the tokens' map spatial resolution.

Following CUT3R (Wang et al., 2025b), we encode past information in a state $s_{t-1} \in \mathbb{R}^{768 \times 768}$. State $s_{t-1}$ and image token $F_t$ interact via two interconnected transformer decoders $\mathcal{D}_F$ and $\mathcal{D}_s$ using cross-attention. This yield, first, image token $F'_t$ enriched with past information from the state $s_{t-1}$ and updated state information $s_t$:

$$[z'_t, F'_t], s_t = \mathcal{D}_F([z, F_t]) \,\circlearrowleft\, \mathcal{D}_s(s_{t-1}) \tag{2}$$

where we define $\mathcal{D}_F \,\circlearrowleft\, \mathcal{D}_s$ as interaction via cross-attention between selected layers of $\mathcal{D}_F$ and $\mathcal{D}_s$.

### 3.3. Disentangling Ego-Motion from World Motion

Prior works on world modeling (Zhou et al., 2025; Karypidis et al., 2025a; Boduljak et al., 2025) focus on forecasting spatial image tokens, thereby modeling ego- and world-motion dynamics induced by the image in a shared (spatial) feature space. In contrast, we propose to additionally learn to operate in the latent space of camera poses. This has two advantages: First, it directly provides all the parameters needed to reconstruct a dynamic 3D environment from only images as input (assuming camera intrinsics remain constant). Second, disentangling pose and spatial latent variables allows us to distinguish between motion induced by the ego-camera and actual 3D scene dynamics.

To handle state-enriched pose $z'_t$ and spatial tokens $F'_t$, we introduce two separate Masked Transformer architectures: a Pose Masked Transformer $M_z$ and a Spatial Masked Transformer $M_F$. $M_z$ forecasts the most likely next pose token and $M_F$ the corresponding spatial tokens. Then these predicted tokens are decoded using pre-trained heads to predict intrinsics, camera pose, and multi-view-consistent depth.

While it is possible to learn forecasting of pose and depth independently, we found that sharing information between $M_z$ and $M_F$ mutually benefits both (see Table 3). This observation is intuitively plausible: On the one hand, reasoning about the next pose by extrapolating the ego-camera motion from the past timesteps simplifies the forecasting of depth regarding static scene structure. This is because camera translation and rotation fully determine how the projected, static scene structure changes from time step $t$ to $t + 1$, assuming both time steps share the same static visual content without occlusions. This is mostly the case for smooth camera trajectories, such as for autonomous driving, except for dynamic regions. On the other hand, reasoning about the next depth prediction directly constrains where the observing camera can be located. More specifically, depth predicted in the camera frame at time $t + 1$ directly induces

the distance between the camera and the scene geometry, and from which view it is observed. Therefore, we get our forecasted 3D scene tokens by:

$$[z'_{t_{N+1}}, F'_{t_{N+1}}] =$$
$$M_z([z'_{t_1}, ..., z'_{t_N}]) \circlearrowleft M_F([F'_{t_1}, ..., F'_{t_N}]) \tag{3}$$

In summary, by allowing information sharing between $M_z$ and $M_F$, we implicitly couple the pose and depth forecasting tasks, thereby enforcing their interdependence and constraining the space of possible solutions.

### 3.4. Efficient Training Strategy & Inference

**Training on latent space** Our main training objective is to learn how to operate on the latent space of a frozen, feed-forward 3D reconstruction model. To achieve this, we employ a teacher-student-distillation strategy, i.e., we leverage our pre-trained 3D reconstruction model as a teacher to train our student forecasting model. As shown in Figure 2, given an image sequence of length $N + 1$, we pre-compute state-enriched 3D scene tokens using our teacher model at every timestep. Meanwhile, we use the first $N$ state-enriched tokens as input to our models $M_z$ and $M_F$ to forecast the next most-likely 3D scene tokens. Inspired by Karypidis et al. (2025a), we formulate the task of future 3D scene prediction as a continuous regression problem. We apply a smooth L1 loss $\ell$ between predicted 3D scene tokens of FR3D (student) $[z'_{t_{N+1}}, F'_{t_{N+1}}]$ and the frozen 3D reconstruction model (teacher), $[\tilde{z}'_{t_{N+1}}, \tilde{F}'_{t_{N+1}}]$, yielding:

$$\mathcal{L}_{\text{pose}} = \mathbb{E}_{s \sim \mathcal{S}} \left[ \ell\left( \tilde{z}'_{t_{N+1}}, \; z'_{t_{N+1}} \right) \right] \text{ and} \tag{4}$$

$$\mathcal{L}_{\text{spatial}} = \mathbb{E}_{s \sim \mathcal{S}} \left[ \frac{1}{|\Omega|} \sum_{p \in \Omega} \ell\left( \tilde{F}'_{t_{N+1}}(p), \; F'_{t_{N+1}}(p) \right) \right] \tag{5}$$

where $\mathcal{S}$ denotes our training sequences and $\Omega$ denotes the set of spatial token locations.

Our final loss is a weighted sum of Equations 4 and 5:

$$\mathcal{L} = \mathcal{L}_{\text{spatial}} + \lambda \, \mathcal{L}_{\text{pose}} \tag{6}$$

where $\lambda = 10$. It is treated as a fixed hyperparameter chosen to balance the relative scale of the two loss terms; we do not claim optimality with respect to $\lambda$.

**Auto-regressive token prediction** In addition, we propose an autoregressive training paradigm. More specifically, our context 3D scene tokens do not only encompass those generated by the teacher but also those from the student. Implementation-wise, we use a sliding-window approach that gradually increases the number of scene tokens predicted by the student relative to those from the teacher,

while keeping the context length $N_c = 4$. This strategy benefits FR3D in two ways: First, we further robustify our student model to forecast plausible next 3D scene tokens despite providing auto-regressive, self-predicted, noisy scene tokens as input; thereby improving FR3D's performance on longer contexts and aligning the training and test setups. Second, we artificially expand the training dataset to ensure our student model can handle longer token sequences, reducing drift in its predictions.

**Inference** After training our pose and spatial forecasting models, FR3D can temporally operate on the latent space of CUT3R (Wang et al., 2025b). In contrast to prior works that rely on dataset-specific decoders (Karypidis et al., 2025a; Boduljak et al., 2025), our frameworks' main advantage is that we can leverage CUT3R's full pipeline, i.e., encoder, decoders, and heads, which are pre-trained for approximately a month on 32 datasets, thereby inheriting its strong generalization capability.

For the inference, we (1) extract state-enriched CUT3R 3D scene tokens from all context input images. (2) the 3D scene tokens (i.e., pose+spatial) are passed to their respective masked transformer to forecast the next most-likely state-enriched token. We (3) follow an autoregressive rollout strategy. As we learn the distribution of state-enriched CUT3R 3D scene tokens directly, we do not need to update CUT3R's state at inference time. The tokens implicitly contain state information.

## 4. Experiments & Results

### 4.1. Experimental Setup

#### 4.1.1. DATASETS & EVALUATION METRICS

For training FR3D, we used the Waymo Open Dataset (Sun et al., 2020), as it is part of our oracle's (i.e., CUT3R's (Wang et al., 2025b)) training distribution, thereby delivering reliable supervision for our student model FR3D. We evaluate our approach in a zero-shot manner on the KITTI (Geiger et al., 2013) and nuScenes (Caesar et al., 2020) datasets. Notably, these two datasets are out-of-training distribution for FR3D and our oracle (see CUT3R's appendix Table 6 (Wang et al., 2025b)). We report standard depth-estimation metrics ($\text{AbsR}$ and $\delta_1$) up to 80m across all datasets, using per-frame scale-and-shift alignment of predictions to the ground-truth. We also report depth evaluation using ground-truth median scaling in the supplementary material, showing the same trend. For pose estimation, we report Absolute Translation Error (ATE), Relative Translation Error ($\text{RPE}_t$), and Relative Rotation Error ($\text{RPE}_R$) after applying Sim(3) alignment with the ground truth, as in CUT3R (Wang et al., 2025b). More specifically, for aligning the poses, we always consider the last context frame as our trajectory's starting point and forecast at least two

*Table 1.* **Zero-shot depth estimation comparison** on KITTI (Geiger et al., 2013) and nuScenes (Caesar et al., 2020). We evaluate KITTI at a resolution of $512 \times 144$ and nuScenes at $512 \times 288$.

| | KITTI | | | | | | nuScenes | | | | | |
| | T+0.6s | | T+1.0s | | T+2.0s | | T+0.75s | | T+1.25s | | T+2.5s | |
| METHOD | AbsR ↓ | $\delta_1$ ↑ | AbsR ↓ | $\delta_1$ ↑ | AbsR ↓ | $\delta_1$ | AbsR ↓ | $\delta_1$ ↑ | AbsR ↓ | $\delta_1$ ↑ | AbsR ↓ | $\delta_1$ ↑ |
|---|---|---|---|---|---|---|---|---|---|---|---|---|
| CUT3R (Oracle) | 0.088 | 0.916 | 0.087 | 0.918 | 0.086 | 0.921 | 0.161 | 0.777 | 0.162 | 0.776 | 0.163 | 0.771 |
| Copy Last | 0.141 | 0.825 | 0.163 | 0.789 | 0.190 | 0.742 | 0.201 | 0.710 | 0.218 | 0.685 | 0.242 | 0.650 |
| DINO-Foresight | 0.128 | 0.857 | 0.156 | 0.807 | 0.197 | 0.745 | 0.223 | 0.705 | 0.242 | 0.673 | 0.283 | 0.609 |
| CUT3R-Foresight | 0.131 | 0.847 | 0.157 | 0.756 | 0.200 | 0.725 | 0.185 | 0.737 | 0.205 | 0.701 | 0.249 | 0.638 |
| **FR3D** | **0.116** | **0.868** | **0.132** | **0.835** | **0.178** | **0.758** | **0.182** | **0.740** | **0.197** | **0.710** | **0.229** | **0.660** |

*Table 2.* **Zero-shot pose estimation comparison** on KITTI (Geiger et al., 2013) and nuScenes (Caesar et al., 2020). We evaluate KITTI at a resolution of $512 \times 144$ and nuScenes at $512 \times 288$.

| | KITTI | | | | | | nuScenes | | | | | |
| | T+1.0s | | | T+2.0s | | | T+1.25s | | | T+2.5s | | |
| METHOD | ATE ↓ | $RPE_t$ ↓ | $RPE_R$ ↓ | ATE ↓ | $RPE_t$ ↓ | $RPE_R$ ↓ | ATE ↓ | $RPE_t$ ↓ | $RPE_R$ ↓ | ATE ↓ | $RPE_t$ ↓ | $RPE_R$ ↓ |
|---|---|---|---|---|---|---|---|---|---|---|---|---|
| CUT3R (Oracle) | 0.066 | 0.122 | 0.236 | 0.119 | 0.146 | 0.251 | 0.098 | 0.150 | 0.254 | 0.226 | 0.219 | 0.273 |
| CUT3R-Foresight + $M_z$ | 0.424 | 0.493 | 1.128 | 0.626 | 0.482 | 1.262 | 0.459 | 0.541 | 0.791 | 0.812 | 0.608 | 0.997 |
| **FR3D** | **0.256** | **0.340** | **0.729** | **0.403** | **0.306** | **0.857** | **0.192** | **0.262** | **0.616** | **0.437** | **0.335** | **0.660** |

additional camera poses to yield a meaningful alignment (since two points connected by a line always result in perfect alignment). Furthermore, to showcase FR3D's abilities in dynamic **indoor** settings, we fine-tuned a model on the Dynamic-RE10K (Chen et al., 2026) (see Table 16 of the supplementary material).

### 4.1.2. BASELINES

Since the task studied in this work is newly defined, there are few methods that directly address the same problem setting. We therefore select baselines from closely related tasks: First, we report our oracle CUT3R (Wang et al., 2025b) and Copy Last as our upper and lower bounds, respectively. Copy Last forecasts depth for subsequent timesteps based on the last observed 3D scene tokens. In addition, we report DINO-Foresight (Karypidis et al., 2025a), a recent work tackling future depth, segmentation, and normals prediction. For fair comparisons, we removed the PCA down-projection module to utilize the full feature resolution of DINOv2 (Oquab et al., 2024) and formulated the depth estimation downstream task as a standard regression rather than a classification problem. We train DINO-Foresight on the same training set as our method. Lastly, we created CUT3R-Foresight as an additional baseline, replacing DINOv2 (Oquab et al., 2024) image features with CUT3R scene tokens.

### 4.1.3. IMPLEMENTATION DETAILS

All models are implemented in PyTorch. Our world model consists of a spatial transformer and a pose transformer with cross-attention-based information sharing. The spatial transformer uses 12 layers with 8 attention heads and a hidden dimension of 1152, operating on high-resolution grid features of size $21 \times 32 \times 3328$. The pose transformer comprises 4 layers with 4 attention heads and a hidden dimension of 1152, processing pose tokens of dimension $1 \times 1 \times 768$. We insert four cross-attention layers between the two transformers, equally spaced throughout the network depth. Both transformers build upon the implementation of DINO-Foresight (Karypidis et al., 2025a) and (Besnier & Chen, 2023). As our representative 3D reconstruction model, we chose CUT3R (Wang et al., 2025b). We train FR3D autoregressively with a fixed context length of 4 frames and an increasing rollout horizon during training, capped at a 5-step prediction. For computational efficiency, features extracted by the CUT3R backbone are precomputed and cached for all training sequences. Optimization uses AdamW with $\beta_1 = 0.9$, $\beta_2 = 0.99$. We use an effective batch size of 32 across 8 NVIDIA A100 GPUs for most training runs. The learning rate is set to $1 \times 10^{-4}$ during pretraining and $5 \times 10^{-5}$ during finetuning, both with cosine annealing schedules. Training proceeds for several pretraining epochs, followed by a small number of finetuning epochs. The training objective is a SmoothL1 reconstruction loss with $\beta = 0.1$.

## 4.2. Quantitative Results

### 4.2.1. ZERO-SHOT EVALUATION

Tables 1 and 2 report the zero-shot performance of FR3D on depth and pose, respectively, on both KITTI (Geiger et al., 2013) and nuScenes (Caesar et al., 2020). Across

*Table 3.* **Ablation study** of our approach FR3D's key components on the Waymo Open Dataset (Sun et al., 2020). A0-A6 use a resolution of $224 \times 224$, while A7-A9 use $512 \times 336$.

| | | DEPTH | | | | CAMERA POSE | | | | | |
| | | T+1.0s | | T+2.0s | | T+1.0s | | | T+2.0s | | |
| ID | METHOD | AbsR ↓ | $\delta_1$ ↑ | AbsR ↓ | $\delta_1$ ↑ | ATE ↓ | RPE$_t$ ↓ | RPE$_R$ ↓ | ATE ↓ | RPE$_t$ ↓ | RPE$_R$ ↓ |
|---|---|---|---|---|---|---|---|---|---|---|---|
| A0 | CUT3R (Oracle) @224 | 0.136 | 0.824 | 0.132 | 0.829 | 0.137 | 0.232 | 0.303 | 0.251 | 0.279 | 0.322 |
| A1 | Copy Last @224 | 0.189 | 0.726 | 0.21 | 0.688 | - | - | - | - | - | - |
| A2 | CUT3R-Foresight @224 | 0.158 | 0.782 | 0.183 | 0.735 | - | - | - | - | - | - |
| A3 | A2 + pose model $M_z$ | 0.158 | 0.782 | 0.183 | 0.735 | 0.489 | 0.570 | 0.507 | 0.895 | 0.654 | 0.724 |
| A4 | A3 + autoregressive | 0.156 | 0.783 | 0.179 | 0.741 | 0.405 | 0.475 | 0.636 | 0.670 | 0.499 | 0.988 |
| A5 | A3 + info share | 0.150 | 0.795 | 0.173 | 0.755 | 0.223 | **0.312** | 0.410 | **0.389** | 0.346 | 0.431 |
| A6 | A4 + A5 = **FR3D** @224 | **0.142** | **0.804** | **0.158** | **0.777** | **0.219** | 0.327 | **0.403** | 0.442 | **0.338** | **0.398** |
| A7 | CUT3R (Oracle) @512 | 0.105 | 0.887 | 0.104 | 0.889 | 0.085 | 0.131 | 0.201 | 0.168 | 0.169 | 0.209 |
| A8 | Copy Last @512 | 0.171 | 0.771 | 0.196 | 0.723 | - | - | - | - | - | - |
| A9 | **FR3D** @512 | **0.131** | **0.840** | **0.152** | **0.800** | **0.173** | **0.241** | **0.356** | **0.532** | **0.368** | **0.388** |

both outputs and all timestamps, the proposed FR3D delivers strong results, also at long time horizons (2 seconds and beyond), surpassing all baselines. Compared to DINO-Foresight (Karypidis et al., 2025a), our model generalizes better thanks to how we leverage the off-the-shelf foundation model. For reference, we first report the performance of our oracle model CUT3R (Wang et al., 2025b), which serves as an upper bound as it has access to all images (i.e., context and future frames) of the evaluated sequence. In addition, we show the performance of the Copy Last baseline and our CUT3R-Foresight, which replaces DINOv2 (Oquab et al., 2024) with CUT3R in DINO-Foresight.

### 4.2.2. COMPONENT ABLATION ON WAYMO

To assess the impact of individual design choices, we perform an ablation study on the Waymo Open Dataset (Sun et al., 2020), shown in Table 3. We report the performance for both forecasting depth and camera pose 1 and 2 seconds ahead. Additional time steps are shown in supplementary material A.2. Methods A0–A6 are trained and evaluated using an input resolution of $224 \times 224$, while methods A7-A9 are subsequently fine-tuned on $512 \times 336$, which yields superior results across the board.

As before, we report our teacher model CUT3R (Wang et al., 2025b) (A0) and the Copy Last baseline (A1) as upper and lower bounds, respectively. Especially for longer horizons of 1 and 2 seconds, A1 performs poorly on depth forecasting due to its naive prediction strategy of reusing the last observed 3D scene tokens for all future time steps.

We started from a modified version of DINO-Foresight (Karypidis et al., 2025a): First, to simplify our training setup, we removed the PCA module, thereby avoiding information loss through feature compression (Boduljak et al., 2025); second, we adapted their architecture to

operate on the multi-scale state-enriched 3D scene token of our reconstruction model, allowing us to reuse its strongly generalizable decoder heads to predict depth. We define this method as CUT3R-Foresight (A2). By learning to forecast how spatial tokens change over time, A2 already shows a significant improvement over A1. Though better than A1, A2 still struggles to forecast depth at longer time horizons, especially 2 seconds ahead.

To additionally predict ego-camera motion, we add our Pose Masked Transformer $M_z$ (A3). A3 can reason about both depth and ego-camera motion, thereby enabling it to tackle future 3D scene reconstruction. However, the results for depth and pose forecasting, especially at a 2-second prediction horizon, remain modest on this early baseline. Notably, the Spatial Masked Transformer and Pose Masked Transformer still predict depth and pose independently.

A4 introduces our autoregressive training strategy. For depth, this yields a small but consistent improvement that tends to increase with longer time horizons. For pose forecasting, despite a higher rotational error compared to A3, the gains in ATE and translational RPE indicate that the component primarily benefits position estimation.

A5 enables information sharing between the Spatial and Pose Masked Transformer via cross-attention on top of A3 to enforce interdependence between both prediction tasks. As a result, A5 yields strong future prediction performance, significantly outperforming methods A1 - A4.

Finally, by combining strategies A4 and A5, we yield FR3D (A6), achieving the strongest performance for future depth prediction and competitive performance for forecasting ego-camera poses compared to A5. Especially for predictions 2 seconds into the future, FR3D achieves significant performance gains over A5, confirming that both information sharing and autoregressive training individually contribute

*Table 4.* **Quantitative depth evaluation** on the Waymo Open Dataset (Sun et al., 2020). All use a resolution of $512 \times 336$.

| | T+1.0s | | T+2.0s | |
|---|---|---|---|---|
| METHOD | AbsR ↓ | $\delta_1$ ↑ | AbsR ↓ | $\delta_1$ ↑ |
| CUT3R-Prompt | 0.198 | 0.690 | 0.228 | 0.620 |
| DINO-Foresight | 0.155 | 0.800 | 0.232 | 0.678 |
| **FR3D** | **0.131** | **0.840** | **0.152** | **0.800** |

to our final method.

### 4.2.3. QUANTITATIVE EVALUATION ON WAYMO

Table 4 reports the quantitative depth comparison of FR3D against two additional baselines on the Waymo Open Dataset (Sun et al., 2020): DINO-Foresight (Karypidis et al., 2025a) and CUT3R-Prompt. CUT3R-Prompt utilizes CUT3R's state readout mechanism to estimate the point map based on GT poses provided as input. As shown, FR3D outperforms both of them. Please note that CUT3R-Prompt relies on GT poses and that its state readout mechanism was trained on 32 datasets, while ours inherits CUT3R's generalization capability, only training on Waymo, highlighting our training strategy's effectiveness.

### 4.2.4. DISENTANGLEMENT ANALYSIS ON WAYMO

We conducted two additional experiments to further investigate FR3D's disentanglement. We define disentanglement as separating motion observed from the visual space (e.g., images) into two distinct yet interconnected sources, ego- (camera) and world- (scene) components.

*Table 5.* **Ablation on the disentanglement** of environment dynamics from the camera motion measuring relative depth change for static and dynamic regions respectively. The "Alignment" column indicates whether frame-wise scale alignment was applied.

| Region | Alignment | T+0.4s | T+0.6s | T+1.0s |
|---|---|---|---|---|
| static | ✗ | 1.32 | 1.87 | 6.73 |
| | ✓ | 0.40 | 0.63 | 1.64 |
| dynamic | ✗ | 8.41 | 12.55 | 17.82 |
| | ✓ | 7.64 | 11.43 | 12.97 |

First, we filter the Waymo Open Dataset (Sun et al., 2020) to obtain a subset of scenes without ego-camera motion. This fixed-camera setting allows us to analyze whether predicted environment dynamics are disentangled from camera motion. Specifically, we measure the relative depth change between a source depth frame $D_{source}$ and a target depth frame $D_{target}$ using the mean absolute log-ratio as a temporal depth-change metric, reported as a percentage. We set $D_{source} = D_{T+0.2s}$ and $D_{target} \in \{D_{T+0.4s}, D_{T+0.6s}, D_{T+1.0s}\}$, and report results in Table 5 for static and dynamic regions,

both with and without frame-wise scale alignment. Static and dynamic regions are separated using the motion mask $M = M_{source} \cup M_{target}$, where $M_{source}$ and $M_{target}$ denote the source and target ground-truth motion masks, respectively.

We observe a drift when forecasting at longer horizons (see also Table 14 in the supplementary material). This can be seen with static regions changing over time and with the relative change between source and target increasing over time. By aligning the frame-wise scale, the drift in static regions between source and target frame decreases to 0.4 - 1.6%. Intuitively, frame-wise alignment reduces the relative depth change for dynamic regions less than for static regions. In summary, regardless of the scale alignment, static regions in FR3D's depth predictions generally expose much less relative change (in %) than dynamic regions.

*Table 6.* **Ablation on the disentanglement** of camera pose prediction accuracy encountering static and dynamic scenes.

| Metric | Scene Motion | T+0.6s | T+1.0s | T+2.0s |
|---|---|---|---|---|
| ATE | static | 0.05 | 0.17 | 0.59 |
| | dynamic | 0.07 | 0.17 | 0.51 |
| $RPE_t$ | static | 0.12 | 0.24 | 0.40 |
| | dynamic | 0.13 | 0.24 | 0.35 |
| $RPE_R$ | static | 0.50 | 0.50 | 0.58 |
| | dynamic | 0.32 | 0.30 | 0.30 |

We further report pose accuracy for static and dynamic scenes on Waymo, respectively. Table 6 shows that FR3D's predicted poses are motion-robust: errors are consistent across static and dynamic scenes through 1s and slightly improve for dynamics at 2s.

### 4.3. Qualitative Results

In the qualitative results, we show the context frames as RGB and the 3D reconstruction produced by our model and the oracle. We indicate the ego-camera poses estimated by the oracle in green, both for the context and the future frames. Instead, the forecasted ego trajectory is indicated in blue. As an upper bound, we show the 3D reconstruction performance of the oracle, given the future input images.

Figure 3 shows qualitative zero-shot results of FR3D on nuScenes (Caesar et al., 2020) and KITTI (Geiger et al., 2013), with two different challenging dynamic scenes for each dataset. Significant domain shifts occur between Waymo (where the FR3D forecasting model is trained) and the two datasets here. For example, the aspect ratio of the KITTI inputs is significantly wider than that of Waymo (Figure 1). Yet, all scenes show remarkable forecasting performance by FR3D, closely matching its oracle model. The scene at the top left depicts a particularly interesting scenario, where a white car is performing a right turn, while

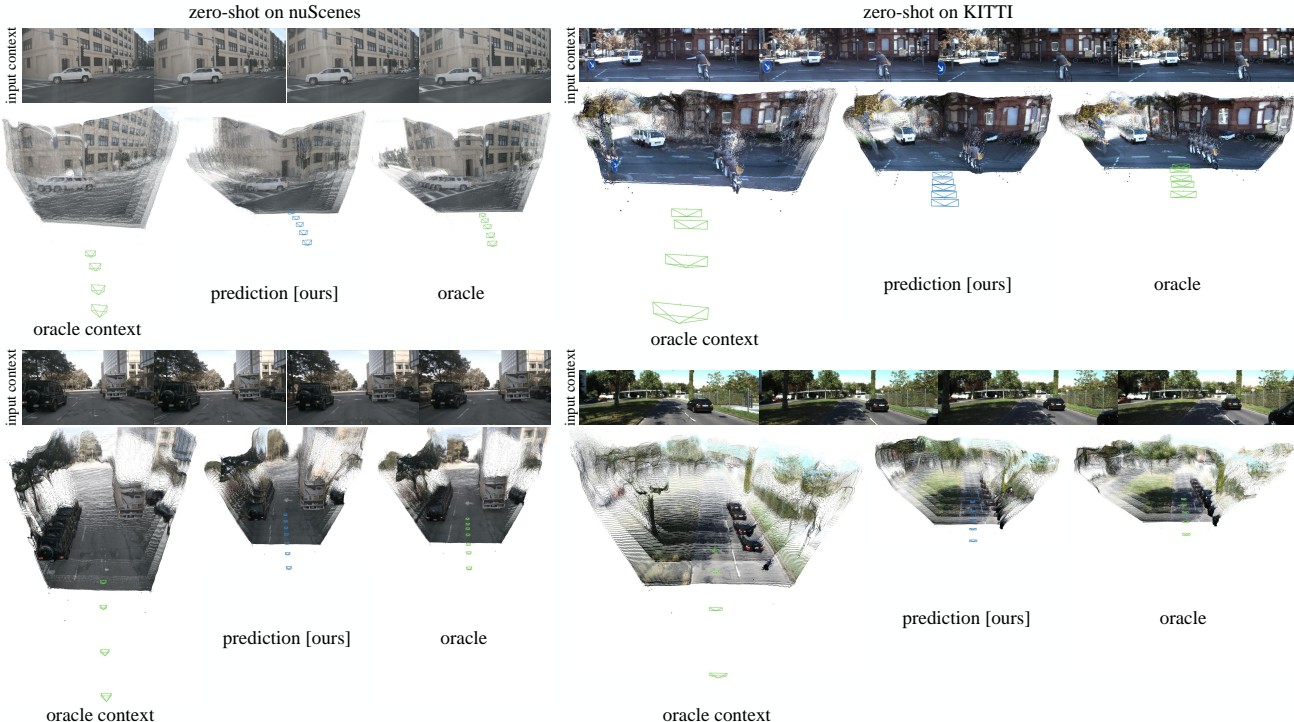

*Figure 3.* Qualitative results on challenging zero-shot dynamic scenes from nuScenes (Caesar et al., 2020) and KITTI (Geiger et al., 2013).

the ego vehicle is turning left onto the same street. FR3D successfully forecasts the ego-motion and the trajectory of the other traffic participant, despite being conditioned only on 4 context images. This shows the benefits of our disentanglement between ego-motion and world-motion, with our model handling the challenging motions independently. The effectiveness of this strategy is also visible in the Waymo (Sun et al., 2020) scene at the top of Figure 1, with traffic plausibly forecasted to head in opposite directions and at different speeds.

The KITTI scene at the top right of Figure 3 also shows an interesting example where a bicyclist is turning left, and the ego vehicle is making a slight left turn, too. Remarkably, the bicyclist's motion is forecasted plausibly despite its slim profile, while simultaneously predicting a plausible ego-motion trajectory to smoothly follow the white van. Furthermore, Figure 3 shows a black vehicle overtaking on the left and a nearly stationary truck (bottom left scene) and two vehicles following one another (bottom right scene). Figure 1 shows a left turn scenario with our model estimating a long, smooth trajectory across the rollout to follow the curve. Comparing this estimate with the oracle trajectory, it can be seen that the oracle's estimates are jittery, introducing significant noise into the training signal of FR3D. However, our autoregressive training helps FR3D be more robust to noisy tokens. Since our forecasting model does not predict RGB, we augment the point clouds with the RGB frames.

## 5. Conclusion

In this work, we introduced FR3D, a 3D world model that predicts the future 3D reconstruction of a scene given a sequence of inputs. By disentangling ego-motion from scene dynamics within a common latent space, our model maintains geometric integrity even far into the future (e.g., 2 seconds). Thanks to our teacher-student strategy leveraging an off-the-shelf foundation model, FR3D shows robust zero-shot generalization without requiring large-scale training. Ultimately, FR3D introduces a new paradigm for world models to operate in the dynamic physical world.

We refer to the **supplementary material** for additional results **indoors**, on static and dynamic regions, other time horizons, depth evaluation using ground-truth median scaling, further geometric analyses, failure cases, and details on runtime and efficiency.

**Limitations** Due to a training data bias on the longitudinal motion of dynamic objects, we notice that our model struggles with objects moving laterally (e.g., cross traffic), exhibiting a tendency to estimate their motion as a mix of lateral and longitudinal motion. We show an example of this in Figure 5 of the supplementary material. This could be addressed by training on more diverse data. Additionally, we observe scale drift in FR3D's predictions, which worsens with each rollout. We believe that additional geometric constraints could improve the predictions.

## Acknowledgements

This project was supported by the BMW Group. In addition, Stefano Gasperini was supported by the German Federal Ministry of Research, Technology and Space (BMFTR) through the Robotics Institute Germany (RIG).

## Impact Statement

This paper introduces future 3D scene prediction and reconstruction, a new 3D computer vision task that can be relevant for safety-critical applications such as autonomous navigation and real-world robotics. To tackle this task, we propose FR3D, a framework for 3D world modeling predicting a persistent 3D latent representation for future dynamic 3D reconstruction by effectively disentangling ego-motion and world-motion. We introduce a teacher-student distillation strategy to bring the up-to-present predictions of a pre-trained 3D reconstruction model to the future by learning to operate on its latent space.

Despite a really data-efficient training strategy, our approach shows robust zero-shot generalization. Extensive experiments demonstrate FR3D's strong performance for future dynamic 3D reconstruction from monocular observations across multiple datasets, even 2 seconds into the future. This may support applications such as planning, navigation, and real-world understanding.

However, the method can fail under heavy distribution shift, degrade under sensor noise or adversarial inputs, and is prone to long-tail scenarios. In such cases, forecasting future ego-pose and scene structure can suffer from degradation. Furthermore, the method could be misused by overly trusting its forecasting outputs, risking unsafe deployment without more thorough validation, particularly in the described failure modes above.

Our results are validated on three different autonomous driving datasets predicting future pose and depth; however, real-world deployment requires additional safety validation.

To reduce risks, we recommend training and evaluating our method in more diverse environments with challenging motion patterns, implementing fail-safe mechanisms in case of deployment, and further investigating how to integrate uncertainty to model the multi-modal future distribution.

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

# A. Additional Quantitative Results

## A.1. Static and Dynamic Scene Predictions

In Tables 7 and 8, we evaluate FR3D's forecasting performance on static and dynamic regions of the Waymo dataset (Sun et al., 2020) separately. This is interesting as it shows the impact of our disentanglement of ego-motion and world-motion. Thanks to that, static regions are anchored in 3D space, allowing them to be estimated significantly more reliably than in our CUT3R-Foresight baseline, even 2 seconds into the future. In particular, at 2 seconds, our FR3D delivers depth estimates comparable to our CUT3R-Foresight baseline at 1 second. Instead, dynamic regions remain challenging and only marginally improve thanks to FR3D, leaving room for future work. These experiments were done at low resolution, i.e., $224 \times 224$.

*Table 7.* **Quantitative evaluation** of our approach FR3D on depth forecasting for **static** regions on the Waymo Open Dataset (Sun et al., 2020). We forecast depth for the following time horizons: 0.2s, 0.6s, 1.0s and 2.0s. We evaluate at a resolution of $224 \times 224$.

| | T+0.2s | | T+0.6s | | T+1.0s | | T+2.0s | |
| METHOD | AbsR $\downarrow$ | $\delta_1 \uparrow$ | AbsR $\downarrow$ | $\delta_1 \uparrow$ | AbsR $\downarrow$ | $\delta_1 \uparrow$ | AbsR $\downarrow$ | $\delta_1 \uparrow$ |
|---|---|---|---|---|---|---|---|---|
| CUT3R (Oracle) | 0.131 | 0.826 | 0.132 | 0.825 | 0.133 | 0.828 | 0.132 | 0.825 |
| Copy Last | 0.148 | 0.798 | 0.166 | 0.764 | 0.179 | 0.738 | 0.200 | 0.699 |
| CUT3R-Foresight | 0.136 | 0.821 | 0.138 | 0.816 | 0.148 | 0.795 | 0.170 | 0.751 |
| **FR3D** | **0.132** | **0.828** | **0.131** | **0.824** | **0.133** | **0.817** | **0.145** | **0.794** |

*Table 8.* **Quantitative evaluation** of our approach FR3D on depth forecasting for **dynamic** regions on the Waymo Open Dataset (Sun et al., 2020). We forecast depth for the following time horizons: 0.2s, 0.6s, 1.0s and 2.0s. We evaluate at a resolution of $224 \times 224$.

| | T+0.2s | | T+0.6s | | T+1.0s | | T+2.0s | |
| METHOD | AbsR $\downarrow$ | $\delta_1 \uparrow$ | AbsR $\downarrow$ | $\delta_1 \uparrow$ | AbsR $\downarrow$ | $\delta_1 \uparrow$ | AbsR $\downarrow$ | $\delta_1 \uparrow$ |
|---|---|---|---|---|---|---|---|---|
| CUT3R (Oracle) | 0.165 | 0.778 | 0.165 | 0.778 | 0.164 | 0.780 | 0.164 | 0.782 |
| Copy Last | 0.216 | 0.708 | 0.290 | 0.611 | 0.329 | 0.563 | 0.365 | 0.512 |
| CUT3R-Foresight | 0.177 | 0.760 | 0.229 | 0.688 | 0.285 | 0.612 | 0.353 | 0.521 |
| **FR3D** | **0.176** | **0.761** | **0.220** | **0.698** | **0.264** | **0.636** | **0.337** | **0.542** |

## A.2. Different Time Horizons

*Table 9.* **Quantitative evaluation** of our approach FR3D on shorter time horizons on the Waymo Open Dataset (Sun et al., 2020). We report 0.2s and 0.6s as well as 0.6s in the future for depth and pose evaluation, respectively.

| | DEPTH | | | | CAMERA POSE | | |
| | T+0.2s | | T+0.6s | | T+0.6s | | |
| METHOD | AbsR $\downarrow$ | $\delta_1 \uparrow$ | AbsR $\downarrow$ | $\delta_1 \uparrow$ | ATE $\downarrow$ | RPE$_t \downarrow$ | RPE$_R \downarrow$ |
|---|---|---|---|---|---|---|---|
| Oracle @224 | 0.140 | 0.817 | 0.137 | 0.82 | 0.092 | 0.204 | 0.292 |
| Copy Last @224 | 0.153 | 0.792 | 0.175 | 0.751 | - | - | - |
| CUT3R-Foresight + $M_z$ @224 | 0.138 | 0.817 | 0.144 | 0.807 | 0.284 | 0.471 | 0.439 |
| **FR3D @224** | **0.135** | **0.823** | **0.137** | **0.815** | **0.091** | **0.190** | **0.435** |
| Oracle @512 | 0.107 | 0.884 | 0.106 | 0.886 | 0.054 | 0.111 | 0.196 |
| Copy-Last @512 | 0.126 | 0.851 | 0.154 | 0.802 | - | - | - |
| **FR3D@512** | **0.110** | **0.879** | **0.121** | **0.858** | **0.065** | **0.131** | **0.370** |

We report additional time horizons for forecasting depth and pose in our zero-shot setting on KITTI (Geiger et al., 2013) and nuScenes (Caesar et al., 2020). These results align with those in the main paper, with the baseline degrading faster than our

model over larger times.

## A.3. Depth Evaluation using Ground-Truth Median Scaling

*Table 10.* **Zero-shot depth comparison** on KITTI (Geiger et al., 2013) and nuScenes (Caesar et al., 2020). We evaluate KITTI at a resolution of $512 \times 144$ and nuScenes at $512 \times 288$ using ground-truth median scaling.

| | KITTI | | | | | | nuScenes | | | | | |
|---|---|---|---|---|---|---|---|---|---|---|---|---|
| | T+0.6s | | T+1.0s | | T+2.0s | | T+0.75s | | T+1.25s | | T+2.5s | |
| METHOD | AbsR ↓ | $\delta_1$ ↑ | AbsR ↓ | $\delta_1$ ↑ | AbsR ↓ | $\delta_1$ | AbsR ↓ | $\delta_1$ ↑ | AbsR ↓ | $\delta_1$ ↑ | AbsR ↓ | $\delta_1$ ↑ |
| CUT3R (Oracle) | 0.090 | 0.911 | 0.090 | 0.913 | 0.089 | 0.918 | 0.153 | 0.781 | 0.153 | 0.780 | 0.156 | 0.774 |
| Copy Last | 0.150 | 0.815 | 0.176 | 0.772 | 0.206 | 0.724 | 0.202 | 0.696 | 0.221 | 0.666 | 0.251 | 0.623 |
| DINO-Foresight | 0.136 | 0.843 | 0.170 | 0.775 | 0.229 | 0.688 | 0.244 | 0.649 | 0.269 | 0.601 | 0.348 | 0.498 |
| CUT3R-Foresight | 0.134 | 0.843 | 0.161 | 0.793 | 0.208 | 0.709 | 0.180 | 0.729 | 0.206 | 0.683 | 0.253 | 0.614 |
| **FR3D** | **0.122** | **0.863** | **0.142** | **0.832** | **0.195** | **0.738** | **0.176** | **0.734** | **0.193** | **0.703** | **0.227** | **0.645** |

*Table 11.* **Depth ablation study** of our approach FR3D on the Waymo Open Dataset (Sun et al., 2020) using ground-truth median scaling. A0-A6 use a resolution of $224 \times 224$ while A7-A9 use $512 \times 336$.

| | | T+0.2s | | T+0.6s | | T+1.0s | | T+2.0s | |
|---|---|---|---|---|---|---|---|---|---|
| ID | METHOD | AbsR ↓ | $\delta_1$ ↑ | AbsR ↓ | $\delta_1$ ↑ | AbsR ↓ | $\delta_1$ ↑ | AbsR ↓ | $\delta_1$ ↑ |
| A0 | CUT3R (Oracle) @224 | 0.139 | 0.802 | 0.137 | 0.806 | 0.136 | 0.809 | 0.133 | 0.813 |
| A1 | Copy Last @224 | 0.154 | 0.777 | 0.179 | 0.732 | 0.196 | 0.701 | 0.219 | 0.657 |
| A2/A3 | CUT3R-Foresight @224 | 0.139 | 0.802 | 0.149 | 0.785 | 0.166 | 0.754 | 0.196 | 0.697 |
| A4 | A3 + autoregressive | 0.140 | 0.801 | 0.148 | 0.786 | 0.164 | 0.759 | 0.192 | 0.709 |
| A5 | A3 + info share | 0.139 | 0.803 | 0.145 | **0.794** | 0.159 | 0.770 | 0.188 | 0.722 |
| A6 | A4 + A5 = **FR3D @224** | **0.136** | **0.808** | **0.142** | **0.794** | **0.151** | **0.778** | **0.175** | **0.735** |
| A7 | CUT3R (Oracle) @512 | 0.109 | 0.870 | 0.108 | 0.872 | 0.107 | 0.872 | 0.107 | 0.875 |
| A8 | Copy Last @512 | 0.130 | 0.836 | 0.161 | 0.781 | 0.181 | 0.745 | 0.208 | 0.693 |
| A9 | **FR3D @512** | **0.113** | **0.864** | **0.127** | **0.836** | **0.138** | **0.814** | **0.166** | **0.766** |

*Table 12.* **Quantitative evaluation** of FR3D on depth forecasting for **static** regions on the Waymo Open Dataset (Sun et al., 2020) using ground-truth median scaling. We forecast depth for the time horizons: 0.2s, 0.6s, 1.0s, 2.0s. We evaluate at a resolution of $224 \times 224$.

| | T+0.2s | | T+0.6s | | T+1.0s | | T+2.0s | |
|---|---|---|---|---|---|---|---|---|
| METHOD | AbsR ↓ | $\delta_1$ ↑ | AbsR ↓ | $\delta_1$ ↑ | AbsR ↓ | $\delta_1$ ↑ | AbsR ↓ | $\delta_1$ ↑ |
| CUT3R (Oracle) | 0.137 | 0.805 | 0.135 | 0.809 | 0.133 | 0.812 | 0.131 | 0.817 |
| Copy Last | 0.149 | 0.784 | 0.171 | 0.742 | 0.186 | 0.713 | 0.208 | 0.669 |
| CUT3R-Foresight | 0.136 | 0.807 | 0.143 | 0.794 | 0.157 | 0.766 | 0.185 | 0.712 |
| **FR3D** | **0.132** | **0.813** | **0.136** | **0.802** | **0.143** | **0.789** | **0.164** | **0.750** |

*Table 13.* **Quantitative evaluation** of FR3D on depth forecasting for **dynamic** regions on the Waymo Open Dataset (Sun et al., 2020) using ground-truth median scaling. We forecast depth for the time horizons: 0.2s, 0.6s, 1.0s, 2.0s. We evaluate at a resolution of $224 \times 224$.

| | T+0.2s | | T+0.6s | | T+1.0s | | T+2.0s | |
|---|---|---|---|---|---|---|---|---|
| METHOD | AbsR ↓ | $\delta_1$ ↑ | AbsR ↓ | $\delta_1$ ↑ | AbsR ↓ | $\delta_1$ ↑ | AbsR ↓ | $\delta_1$ ↑ |
| CUT3R (Oracle) | 0.170 | 0.759 | 0.171 | 0.759 | 0.167 | 0.765 | 0.167 | 0.767 |
| Copy Last | 0.220 | 0.690 | 0.297 | 0.590 | 0.334 | 0.540 | 0.369 | 0.490 |
| CUT3R-Foresight | 0.181 | 0.742 | 0.233 | 0.669 | 0.286 | 0.590 | 0.349 | 0.502 |
| **FR3D** | **0.179** | **0.745** | **0.222** | **0.683** | **0.261** | **0.622** | **0.327** | **0.526** |

Table 10, 11, 12, and 13 report depth evaluation similar to Table 1, 3 (left part), 7, and 8 respectively but using ground-truth median scaling to align predicted and ground-truth depth for evaluation. The reported results align with those in the main paper, with the baselines degrading faster than our model over longer time horizons. For all evaluations, we included our oracle CUT3R (Wang et al., 2025b) as reference.

Table 10 evaluates FR3D's zero-shot performance on depth forecasting on the KITTI (Geiger et al., 2013) and nuScenes (Caesar et al., 2020) dataset against three baselines, Copy Last, DINO-Foresight (Karypidis et al., 2025a), and CUT3R-Foresight. FR3D outperforms all of them at all time horizons across both datasets.

Table 11 shows our ablation study regarding depth prediction accuracy on the Waymo Open dataset (Sun et al., 2020). A0 - A6 operate on a resolution of $224 \times 224$, while A7 - A9 use $512 \times 336$. For both resolutions, the Copy Last baseline (A1) performs the worst, strongly degrading with longer time horizons. Our CUT3R-Foresight baseline A2 significantly improves over A1, indicating that CUT3R's spatial features can serve as representation space for depth forecasting. While A4 and A5 yield approximately the same performance for 0.2s as A2, autoregressive training and information sharing between pose and depth especially help for longer time horizons (i.e., 0.6s, 1.0s, and 2.0s) with information sharing being more beneficial. Combining both yields FR3D (A6) showing the strongest depth forecasting performance. Fine-tuning on higher input image resolution leads again to better performance (A9).

Table 12 and 13 evaluate FR3D's depth forecasting performance for static and dynamic regions against Copy Last and CUT3R-Foresight on the Waymo Open dataset (Sun et al., 2020). FR3D outperforms both of them being especially strong for static regions supporting our assumption that sharing information between pose and depth primarily benefits the depth prediction of static scene parts. Although FR3D outperforms the baselines in depth forecasting for dynamic regions, accurately modeling scene dynamics remains challenging.

## B. Additional Qualitative Results

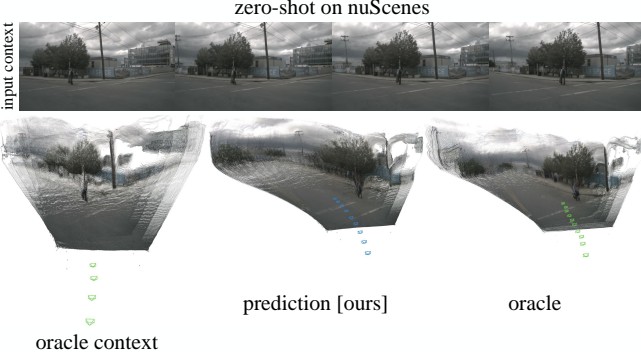

*Figure 4.* Zero-shot prediction of our model on the nuScenes dataset. The example shows a turning scenario.

In Figure 4, we show an interesting qualitative result with a zero-shot prediction of our model on the nuScenes (Caesar et al., 2020) dataset. The scene exhibits a left-hand turn. In the context, it can be seen that the turn has just started (visible in the camera poses). Our model correctly predicts a smooth completion of the turn throughout the rollout (blue cameras). Remarkably, the pose prediction of the oracle (right) is more jittery than that of our model, although our model is trained to mimic the oracle predictions. However, we can explain the improvement as our model learns on noisy tokens thanks to our autoregressive strategy. This makes it robust against the oracle tokens, too.

In Figure 5, we show two failure cases of our model on Waymo scenes. As discussed in the limitations (Section 5), our model has a bias toward longitudinal motion and struggles with lateral motion. As a result, it combines lateral and longitudinal motion when lateral motion occurs. More diverse training data should mitigate this, but it is out of the scope of this work.

## C. Scale Drift

We further investigate to which extend our model is prone to scale drift. Table 14 shows the average relative depth scale, comparing FR3D with its oracle. FR3D shows depth drift, acting as a "persistent 3D world model up to scale" (persistent here means in a coherent/joint coordinate system over time). Considering that FR3D was trained on a single dataset only

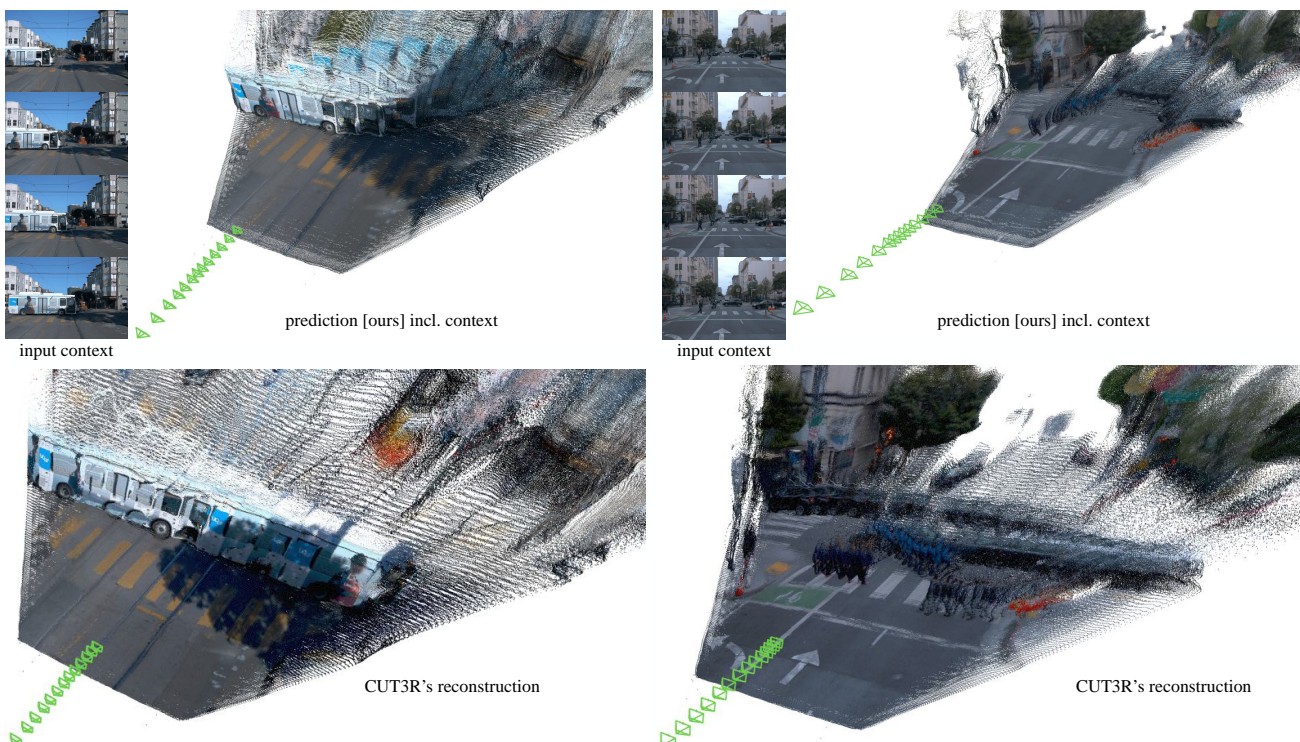

*Figure 5.* Failure cases from two scenes of the Waymo dataset, where our model mixes longitudinal and lateral motion for the dynamic objects. In both cases, the ego vehicle is slowing down approaching an intersection. Context and rollouts are displayed together in the 3D reconstruction. CUT3R's reconstruction on the full sequence is included as reference in the bottom row.

while CUT3R utilized 32 datasets (Wang et al., 2025b), we believe that extended training with more data diversity could mitigate it. Another option could be introducing a scale token to decouple scale prediction from spatial tokens (similar to MapAnything (Keetha et al., 2026)). Please note that CUT3R has access to future frames for depth prediction, while FR3D forecasts depth.

*Table 14.* **Evaluation on scale drift** of our approach FR3D on depth forecasting on the Waymo Open Dataset (Sun et al., 2020). We forecast depth for the following time horizons: 0.2s, 0.6s, 1.0s and 2.0s. We compare against CUT3R's relative depth scale for the respective time steps.

| METHOD | T+0.2s | T+0.6s | T+1.0s | T+2.0s |
|---|---|---|---|---|
| CUT3R (Oracle) | 1.67 | 1.68 | 1.68 | 1.67 |
| FR3D | 1.54 | 1.42 | 1.31 | 1.18 |

## D. Shape Stability

Additionally, we conducted an experiment to assess the shape stability of dynamic objects for 4D consistency evaluation. First, we extract the predicted 3D point cloud of each dynamic object at time and use 4D consistent instance labels from the Waymo val split. Both are transformed into the same coordinate frame by compensating the ego-motion given predicted ego poses. Moreover, we remove object motion, by estimating the rigid transformation between both object point clouds using ICP. Then, we remove monocular scale drift by estimating a single global scale correction from static background pixels. Finally, we measure the remaining difference between both aligned point clouds using symmetric Chamfer distance.

In Table 15 compare the shape stability of CUT3R's and FR3D's dynamic reconstruction. Both methods show very similar performance, with nearly identical Chamfer distances across all time steps. The error remains stable as the temporal gap increases, indicating consistent shape preservation over time.

*Table 15.* **Ablation on shape stability** comparing CUT3R and FR3D on the Waymo val split (Sun et al., 2020) using symmetric Chamfer distance.

| METHOD | T+0.2S → T+0.4S | T+0.4S → T+0.6S | T+0.8S → T+1.0S |
|---|---|---|---|
| CUT3R | 0.148 | 0.143 | 0.146 |
| FR3D | 0.157 | 0.141 | 0.141 |

# E. Beyond Driving

Our paper mainly focuses on the driving domain due to its structured nature. Nevertheless, we conducted an experiment where we fine-tune FR3D on Dynamic-RE10K (Chen et al., 2026), a large-scale real-world, dynamic, indoor dataset. Since Dynamic-RE10K does not provide GT depth and pose, we follow DINO-Foresight and evaluate against our oracle CUT3R (Wang et al., 2025b). We report rollouts (i.e., t+k) in frames rather than seconds for Dynamic-RE10K, since the frame rate varies across scenes. As shown in Table 16, FR3D outperforms the baseline for depth and closely follows CUT3R's trajectory.

*Table 16.* **Quantitative evaluation** on the Dynamic-RE10K dataset (Chen et al., 2026).

| | T+1 | | T+3 | | T+5 | |
|---|---|---|---|---|---|---|
| METHOD | AbsR ↓ | $\delta_1$ ↑ | AbsR ↓ | $\delta_1$ ↑ | AbsR ↓ | $\delta_1$ ↑ |
| Copy Last | 0.03 | 0.98 | 0.08 | 0.91 | 0.12 | 0.86 |
| FR3D (finetuned) | **0.02** | **0.99** | **0.06** | **0.96** | **0.10** | **0.90** |

| | T+3 | | | T+5 | | |
|---|---|---|---|---|---|---|
| METHOD | ATE ↓ | $RPE_t$ ↓ | $RPE_R$ ↓ | ATE ↓ | $RPE_t$ ↓ | $RPE_R$ ↓ |
| FR3D (finetuned) | 0.01 | 0.03 | 0.51 | 0.02 | 0.04 | 0.63 |

# F. Details on the Masked Transformer Architecture

Both the spatial and pose forecasting modules follow a masked-token prediction formulation inspired by DINO-Foresight (Karypidis et al., 2025a). During training, tokens corresponding to future timesteps are replaced by learned mask tokens and reconstructed from context tokens in a single forward pass (i.e., prediction is not iterative MaskGIT-style refinement). The spatial masked transformer operates on a spatio-temporal grid of state-enriched spatial tokens with learned 2D spatial positional encodings and temporal positional encodings, and each layer consists of temporal self-attention across frames followed by spatial self-attention within each frame. The pose masked transformer processes pose tokens using temporal self-attention with learned temporal positional encodings. The two streams are coupled via bidirectional cross-attention layers inserted at multiple depths, allowing spatial tokens to condition on predicted ego-motion and pose tokens to condition on scene geometry.

# G. Runtime & Efficiency

As shown in Table 17, we conducted a profiling of peak VRAM and inference latency comparison of FR3D against Vista (Gao et al., 2024), a recent 2D video world model for driving scenes. We report inference latency when forecasting 2 seconds into the future using 4 context frames.

*Table 17.* **Evaluation of runtime and efficiency** of FR3D compared to Vista (Gao et al., 2024). We report peak VRAM and inference latency.

| METHOD | VRAM (GB) | Latency (s) |
|---|---|---|
| Vista | 11.60 | 24.57 |
| FR3D | 5.75 | 0.85 |

Furthermore, Table 18 shows a detailed profiling of the latency of FR3D's individual key steps.

*Table 18.* **Latency profiling** of FR3D's individual key steps. We report peak VRAM and inference latency.

| MODULE | Latency (s) |
|---|---|
| Context Step (CUT3R) | 0.044 |
| Forecast Step (FR3D) | 0.038 |
| Head Prediction Step (CUT3R or FR3D) | 0.017 |
| Context for 4 frames (CUT3R) | 0.175 |
| Forecast for 10 frames (FR3D) | 0.545 |
| Total (FR3D) | 0.854 |

We used a NVIDIA A100-SXM4-40GB GPU for all profiling experiments.

## H. Notation Table

For clarity, Table 19 summarizes the main symbols used throughout the method section, including the notation for observations, latent 3D scene tokens, forecasting modules, and training objectives.

*Table 19.* Notation used throughout the paper and supplementary material.

| Symbol | Meaning |
|---|---|
| *General world-model notation* | |
| $t$ | Discrete time step |
| $x_t$ | Observation at time $t$ |
| $a_t$ | Action taken at time $t$ |
| $s_t$ | System state at time $t$ |
| $h_t$ | Encoded observation at time $t$ |
| $u_t$ | Latent variable representing unobserved or unknown factors |
| *Input and reconstruction notation* | |
| $I$ | Sequence of context images |
| $I_t$ | Context image at time $t$ |
| $N$ | Number of context images |
| $H, W$ | Input image height and width |
| $R$ | Pre-trained feed-forward 3D reconstruction model |
| $\mathcal{E}$ | Pre-trained image encoder of $R$ |
| $F$ | Sequence of per-frame image tokens |
| $F_t$ | Image tokens at time $t$ |
| $D$ | Image-token feature dimension |
| $H_F, W_F$ | Spatial resolution of the image-token map |
| @ | Input image resolution at which a model is evaluated, e.g. CUT3R (Oracle) @224 |
| *Latent 3D scene forecasting notation* | |
| $\mathcal{D}_F$ | Transformer decoder operating on image tokens |
| $\mathcal{D}_s$ | Transformer decoder operating on state tokens |
| $F'_t$ | State-enriched spatial image tokens at time $t$ |
| $z'_t$ | State-enriched pose token at time $t$ |
| $s_{t-1}, s_t$ | Previous and updated recurrent scene states |
| ♋ | Cross-attention interaction between transformer modules |
| $M_z$ | Pose Masked Transformer |
| $M_F$ | Spatial Masked Transformer |
| $z'_{t_{N+1}}$ | Predicted next pose token |
| $F'_{t_{N+1}}$ | Predicted next spatial tokens |
| $\tilde{z}'_{t_{N+1}}$ | Teacher pose token at time $t_{N+1}$ |
| $\tilde{F}'_{t_{N+1}}$ | Teacher spatial tokens at time $t_{N+1}$ |
| *Training notation* | |
| $\mathcal{S}$ | Set of training sequences |
| $\Omega$ | Set of spatial token locations |
| $p$ | Spatial token location, $p \in \Omega$ |
| $\ell$ | Smooth L1 loss |
| $\mathcal{L}_{\text{pose}}$ | Pose-token distillation loss |
| $\mathcal{L}_{\text{spatial}}$ | Spatial-token distillation loss |
| $\mathcal{L}$ | Final weighted training objective |
| $\lambda$ | Weight of the pose loss term |
| $N_c$ | Context length used for forecasting |
| $\beta_1, \beta_2$ | AdamW momentum coefficients |
| $\beta$ | Smooth L1 threshold at which the loss changes between L1 and L2 behavior |
| *Fixed-camera disentanglement analysis* | |
| $D_{\text{source}}$ | Source depth frame |
| $D_{\text{target}}$ | Target depth frame |
| $D_{t+\Delta}$ | Depth frame at time offset $\Delta$ from $t$ |
| $M$ | Motion mask used to separate static and dynamic regions |
| $M_{\text{source}}$ | Ground-truth motion mask of the source frame |
| $M_{\text{target}}$ | Ground-truth motion mask of the target frame |
| $M_{\text{source}} \cup M_{\text{target}}$ | Union of source and target motion masks |

