# OpenReview forum: "Future Dynamic 3D Reconstruction: A 3D World Model with Disentangled Ego-Motion"
_ICML.cc/2026/Conference — ICML 2026 regular_

### Official Review · Reviewer_oa97 · 2026-02-27

**Soundness:** 2
**Presentation:** 2
**Significance:** 3
**Originality:** 2
**Overall Recommendation:** 4
**Confidence:** 3

**Summary:**

This paper proposes **FR3D**, a world model for forecasting **future dynamic 3D reconstructions** from short monocular video context. The key idea is to operate directly in the **latent/state space of a frozen feed-forward 3D reconstruction foundation model** (CUT3R): FR3D predicts next-step **pose tokens** and **spatial scene tokens** using two masked transformers with cross-attention information sharing. Training uses **teacher–student distillation on token targets** and an **autoregressive schedule** to reduce rollout drift. Results show improved zero-shot depth and pose forecasting up to ~2 seconds.

**Compliance With Llm Reviewing Policy:**

Affirmed.

**Final Justification:**

The paper presents a coherent and well-motivated approach to forecasting in the latent space of a 3D reconstruction model, and the rebuttal provided useful additional experiments on disentanglement and beyond-driving generalization that address some of my initial concerns. While the core claims around principled disentanglement, persistent metric 3D consistency, and independence from teacher biases would benefit from more explicit enforcing mechanisms, weaker-alignment pose evaluations, or teacher generality tests, the overall contribution is solid and the empirical results are convincing. I upgrade my recommendation to weak accept, as the work makes meaningful progress on an important problem, though some claims could be better substantiated with additional evidence.

**Key Questions For Authors:**

1. **Disentanglement claim:** Beyond using separate pose/spatial transformers, is there any explicit mechanism or measurable criterion that enforces ego-motion/world-motion disentanglement? Can you provide a quantitative diagnostic (e.g., performance split on static vs dynamic regions for pose, or sensitivity of pose prediction to dynamic-object motion)?
2. **Teacher data / OOD clarity:** You state KITTI and nuScenes are out-of-training-distribution for both FR3D and the CUT3R teacher. Can you list the datasets used to pretrain CUT3R (or confirm explicitly that KITTI/nuScenes are excluded) to rule out data leakage?
3. **Metric consistency evaluation:** How does FR3D perform if you (a) remove per-frame mean scaling for depth and (b) reduce/remove Sim(3) alignment for pose? Even if absolute scale is ambiguous, can you report scale drift statistics across rollout steps?
4. **Inference loop details:** At test time, how exactly is the CUT3R state updated when only predicted tokens are available? A short pseudocode for the rollout (state update, token prediction, decoding) would clarify how “persistent state” is maintained and where drift may accumulate.

**Limitations:**

yes

**Strengths And Weaknesses:**

**Strengths**
- The methodology is coherent: forecasting in the latent space of a strong 3D reconstruction model (CUT3R) is a reasonable way to inject 3D inductive bias while avoiding pixel-space rollout artifacts.
- The paper provides targeted ablations (Table 3) showing that (i) pose modeling, (ii) autoregressive training, and (iii) pose–spatial information sharing each contribute, with the largest gains coming from information sharing and improved long-horizon behavior.
- The training objective is simple and stable (SmoothL1 token regression), and the teacher–student setup is clearly motivated to leverage a pretrained model’s geometric priors.

**Weaknesses**
- **Uncertainty is not modeled**, despite the preliminaries motivating latent variables for stochasticity. FR3D is effectively a deterministic “most-likely” predictor trained by regression to a single teacher trajectory. In dynamic scenes (e.g., merges, occlusions, multiple agents), the future is inherently multi-modal; a deterministic SmoothL1 objective can encourage averaging behavior, which may manifest as incorrect or implausible motion for dynamic objects. This also limits direct use for decision making under risk (sampling multiple futures).
- The “disentanglement” claim is currently **more architectural than principled**. The model uses separate pose and spatial transformers, but then explicitly **couples them via cross-attention**. That can be beneficial (as the ablation shows), yet it weakens the interpretation of a clean separation between ego-motion and world-motion. There is no explicit identifiability argument or quantitative disentanglement diagnostic (e.g., measuring how pose predictions change under different dynamic-object motion patterns).
- **Strong dependence on the teacher and its biases.** The student is trained to mimic CUT3R’s state-enriched tokens; thus, the ceiling and failure modes are constrained by CUT3R’s representation and supervision quality. The paper acknowledges teacher pose “jitter” qualitatively; it would strengthen soundness to quantify how teacher noise propagates and when the student “denoises” versus introduces bias.
- **Evaluation choices may hide key failure modes for a “persistent 3D world model.”**
  - Depth is evaluated with **per-frame mean scaling**, which removes global scale and can mask temporal scale drift. For a persistent 3D state, scale consistency across time is central.
  - Pose is evaluated after **Sim(3) alignment**, which can also absorb scale inconsistencies. This is standard in some reconstruction settings, but it makes it harder to judge whether FR3D maintains a metrically consistent 3D world over long rollouts.
- **Action conditioning is not truly present.** The paper frames inferred ego-motion as a “proxy for action,” but there is no explicit action input at inference time. As a result, FR3D is primarily a forecaster of future ego-trajectories, not a controllable simulator that can be queried under counterfactual actions. This is not a fatal flaw, but it should be clearly scoped as forecasting rather than control-conditioned world modeling.
- Some details that matter for reproduction and long-horizon reasoning remain underspecified, such as the **exact inference loop and state update** across predicted steps (what precisely is fed back, how the CUT3R state is updated when future images are not available, and whether any normalization/clipping is used to prevent drift).

### Presentation

**Strengths**
- The narrative is generally easy to follow, and the pipeline is well communicated visually (Fig. 2).
- The experimental section is structured well: clear baseline selection for a newly defined task, plus ablations and qualitative results.

**Weaknesses**
- Notation overload can cause avoidable confusion: the paper introduces a generic latent variable in the world-model preliminaries, and later uses similarly named pose tokens; “state” refers to both generic world-model state and CUT3R’s persistent state. A small notation table would help.
- “Masked transformer” components are described at a high level, but important implementation specifics are missing or scattered (masking strategy, positional encoding choices for spatial tokens, and whether prediction is single-pass vs iterative). These details affect whether the model is best interpreted as masked regression, masked autoregressive modeling, or something closer to MaskGIT-style generation.
- The PDF includes a sentence that appears to **instruct reviewers to include specific phrases in their review**. This should be removed immediately; it reads like an artifact at best and a review-manipulation attempt at worst.
- The Impact Statement is extremely generic given the likely use in safety-critical contexts (autonomy/robotics). A short concrete discussion of failure modes and misuse would improve completeness.

### Significance

**Strengths**
- The problem is important: long-horizon rollout consistency is a core gap for current video world models, and 3D consistency is crucial for embodied agents.
- The approach is practical: it leverages an existing 3D reconstruction foundation model and shows strong **zero-shot** transfer to KITTI and nuScenes, which is meaningful for real-world deployment constraints.

**Weaknesses**
- The demonstrated scope is mostly within the **driving domain** (Waymo → KITTI/nuScenes). It is unclear whether the approach generalizes to other settings with different dynamics (indoor humans, articulated/non-rigid objects, handheld camera motion).
- Because it outputs a single “best guess,” the method is less directly impactful for applications that need calibrated uncertainty (planning, safety envelopes).

### Originality

**Strengths**
- Forecasting **both** ego-pose and 3D spatial tokens in the latent/state space of a modern feed-forward 3D recon model is a meaningful conceptual bridge between “reconstruction models” and “world models.”
- The pose–spatial cross-attention design plus autoregressive distillation is a well-motivated combination, and the ablation provides evidence that it is not just cosmetic.

**Weaknesses**
- The learning recipe (masked-token forecasting + regression + teacher distillation) is closely related to recent feature-forecasting world models (e.g., DINO-Foresight-style designs). Much of the novelty appears to come from **retargeting** this machinery to CUT3R’s 3D token/state space and adding a pose stream. The paper would benefit from a sharper novelty statement: what is fundamentally new beyond “apply masked forecasting to a stronger 3D backbone”?

### Experimental strengths / weaknesses and suggested additions

**Strengths**
- Strong zero-shot results on KITTI and nuScenes across multiple horizons (Tables 1–2).
- Useful ablations (Table 3) and additional analysis separating static vs dynamic regions (supplementary Tables 4–5).
- Qualitative results are convincing for long-horizon stability, and the paper is honest that dynamic regions remain harder.

**Weaknesses / important missing checks**
- Add at least one evaluation that probes **temporal and metric consistency** without per-frame depth scaling (e.g., scale drift over time, or frame-to-frame point-cloud consistency using predicted poses).
- Report pose errors with **weaker alignment** (e.g., SE(3) alignment or even no alignment after anchoring at the last context frame) to better reflect absolute drift. If monocular scale ambiguity is the reason, explicitly discuss what “persistent 3D” means under this ambiguity.
- The “framework” claim would be stronger with a minimal test of **teacher/backbone generality** (even a small-scale experiment with another pretrained reconstruction model, or a discussion plus partial results if full retraining is expensive).
- Since dynamic regions are a major failure mode, consider adding an object-centric diagnostic (even simple: dynamic-mask depth error trends vs horizon) and more systematic failure statistics beyond a couple of examples.

---

> ### Author Rebuttal · Authors · 2026-03-31
>
> Thank you for your detailed review, finding our problem “important” and our “narrative [...] easy to follow”, with FR3D as a “meaningful conceptual bridge between reconstruction models and world models” and “well motivated” design choices. Please find our replies by topic here, within the 5k char limit.
> ## 1 Disentanglement
> We define disentanglement as separating observed motion into ego- (camera) and world- (scene) components. We employ dual masked transformers with separate regression losses; cross-attention then models their correlation (e.g., depth change relative to ego-motion). DINO-Foresight and VFMF use a single feature space, but our decoupled streams enable better motion attribution. As suggested, we report pose accuracy for static/dynamic scenes on Waymo. FR3D poses are motion-robust: errors are consistent across static/dynamic scenes through 1s and improve a bit for dynamics at 2s. Our proposed disentanglement technique effectively simplifies forecasting joint ego- and world-motion.
> Metric|Motion|<=t+0.6|<=t+1|<=t+2
> -|-|-|-|-
> ATE|no|0.05|0.17|0.59
> ||yes|0.07|0.17|0.51
> RPE_t|no|0.12|0.24|0.40
> ||yes|0.13|0.24|0.35
> |RPE_R [deg]|no|0.50|0.50|0.58
> ||yes|0.32|0.30|0.30
> ## 2 Beyond driving
> Beyond driving, we fine-tune here on the indoor Dynamic-RE10K [3]. Lacking GT, we evaluate against CUT3R. We report rollouts in frames due to varying FPS. FR3D outperforms the baseline for depth and closely follows CUT3R’s trajectory.
> Depth (AbsR / δ1)|t+1|t+3|t+5
> -|-|-|-
> Copy Last|0.03 / 0.98|0.08 / 0.91|0.12 / 0.86
> FR3D fine-tuned|0.02 / 0.99|0.06 / 0.96|0.10 / 0.90
>
> Pose (ATE / RPE_t / RPE_R)|t+3|t+5
> -|-|-
> FR3D fine-tuned|0.01 / 0.03 / 0.51|0.02 / 0.04 / 0.63
> ## 3 Dynamic errors
> Please see Tables 4 and 5 for static/dynamic.
> ## 4 Inference
> Steps: a) extract state-enriched CUT3R tokens from context; b) forecast the next state-enriched token via masked transformers; and c) autoregressive rollout. Explicit CUT3R state updates are unnecessary as tokens implicitly capture temporal state.
> ## 5 Mask transformers
> Both use a non-iterative masked-token formulation (DINO-Foresight), reconstructing future tokens in one pass. The spatial one interleaves temporal and spatial self-attention over state-enriched tokens (using 2D and temporal encodings). The pose one applies temporal self-attention to pose tokens. We link them via multi-depth bidirectional cross-attention, enabling joint reasoning over ego-motion and scene geometry while maintaining distinct tokens.
> ## 6 Teacher dependency
> Please refer to Reviewer p9ox rebuttal 4.
> ## 7 Teacher jitter
> Autoregressive training reduces CUT3R’s pose jitter, yet rotation degrades a bit (Table 3 A3-4). Quantifying noise propagation is challenging as reconstruction and forecasting involve different tasks and input horizons. Training with injected pose noise would be valid but infeasible due to limited time.
> ## 8 Novelty
> Our novelty is 3-fold: a) we introduce the future 3D scene reconstruction task; b) we forecast the 3D scene by projecting pre-trained teacher predictions into the future; and c) disentangle observed motion into distinct ego- and world-motion components.
> ## 9 CUT3R training data and KITTI/nuScenes in it?
> KITTI/nuScenes are indeed excluded. A full list is in CUT3R’s appendix Table 6 [1].
> ## 10 Injected LLM prompt
> We did not add it, but it is part of ICML’s efforts on LLM usage (see Peer Review FAQ).
> ## 11 Scale drift
> Thanks. We show the avg. relative depth scale here, comparing FR3D with its oracle. FR3D shows depth drift, acting as a “persistent 3D world model up to scale” (persistent here means in a coherent/joint coordinate system over time). Extended training or a scale token to free it from spatial tokens (e.g., MapAnything [2]) could mitigate it.
> ||t+0.2|t+0.6|t+1|t+2
> -|-|-|-|-
> CUT3R|1.67|1.68|1.68|1.67
> FR3D|1.54|1.42|1.31|1.18
> ## 12 Uncertainty and multi-modality
> FR3D focuses on the "most-likely" future, with multi-modal extensions deferred to future work. We will clarify this in the final version. See response 5 to Reviewer p9ox.
> ## 13 Teacher agnostic
> While adaptable to other feed-forward models (e.g., MapAnything [2]), it is infeasible now. Note that unlike permutation-invariant offline works, FR3D requires chronological input.
> ## 14 Impact statement
> Our updated impact statement discusses failure modes (distribution shift, noise, long-tail scenarios) and the risk of over-reliance in safety-critical settings. Mitigation via diverse training and uncertainty modeling is future work.
> ## 15 Notation
> Thanks, we now use u instead of z as the generic latent variable in Section 3.1. Will add a notation table.
> ## 16 FR3D as forecaster
> Thanks, we will clarify it in the final version.
> ## Refs:
> [1] Wang et al., Continuous 3D Perception Model with Persistent State, CVPR 2025.
>
> [2] Keetha et al., MapAnything: Universal Feed-Forward Metric 3D Reconstruction, 3DV 2026.
>
> [3] Chen et al., WildRayZer: Self-supervised Large View Synthesis in Dynamic Environments, CVPR 2026.

---

> > ### Author Rebuttal · Reviewer_oa97 · 2026-04-02
> >
> > Thank you for the detailed rebuttal. I appreciate the additional analyses and clarifications.
> >
> > **Issues that are resolved or largely resolved**
> > - **OOD / data leakage:** You clearly confirm that KITTI and nuScenes are excluded from CUT3R pretraining.
> > - **Framing:** It is now clearer that FR3D is a forecaster of the most-likely future, not an action-conditioned simulator. This addresses that concern mainly at the scope/claim level.
> > - **Presentation:** The notation fix, clearer inference description, and updated impact statement are helpful. Also, the reviewer-instruction sentence appears to be a venue artifact, so I no longer consider it an author-side issue.
> > - **Additional evidence:** The added static vs. dynamic pose analysis and the small indoor experiment are useful additions.
> >
> > **Issues that are still unresolved or only partially resolved**
> > - **Disentanglement:** The new results are helpful, but they still support architectural separation more than principled disentanglement. There is still no explicit mechanism or stronger diagnostic showing that ego-motion prediction is robust to dynamic-object motion.
> > - **Metric consistency:** The new scale analysis is appreciated, but it also shows clear scale drift over time. In addition, pose is still not evaluated under weaker alignment, so the claim of a persistent 3D world remains only partially supported.
> > - **Teacher dependence / noise propagation:** This remains largely unanswered. The rebuttal does not quantify when the student denoises versus inherits teacher bias, and backbone generality is still untested.
> > - **Beyond-driving generalization:** The indoor result is encouraging, but since it uses fine-tuning and teacher-based evaluation rather than GT, it only partially addresses the generalization concern.
> > - **Uncertainty / multimodality:** This is acknowledged as future work, but it remains an important limitation for planning or safety-critical use.
> >
> > **Final judgment**
> >
> > I would keep my recommendation at weak reject and would not support acceptance in the current form. The paper has clear strengths, and the rebuttal improves the framing, but the main technical claims—especially disentanglement, persistent metric 3D consistency, and teacher dependence—are still not validated strongly enough.

---

> > > ### Author Response · Authors · 2026-04-08
> > >
> > > Thank you for acknowledging our rebuttal. Please find below our responses to the partially open aspects.
> > >
> > > ## Disentanglement
> > > We believe there is a misunderstanding about the disentanglement experiment we provided in the rebuttal. Due to the 5k-character limit, we could not explain in more detail. We clarify it here:
> > >
> > > We showed that our model effectively disentangles ego-motion from world-motion as its pose predictions are not affected by the motion in the scene. We showed the pose accuracy on static vs. dynamic scenes (Waymo val.), with ATE, RPE_t and RPE_R up to the specified time step rollout (e.g., "<= t+1.0s" incorporates all rollout up to timestep t+1.0s to compute the metric). As shown, FR3D performs similarly regardless of dynamic object motions, confirming the disentanglement is not only architectural but also experimentally validated.
> > >
> > > We report here the same experiments but showing the difference in the metrics between static and dynamic (as dynamic - static, so negative values mean that the results in the dynamic scenes had lower errors):
> > >
> > > $\Delta$ Metric between static and dynamic|<=t+0.6|<=t+1|<=t+2
> > > -|-|-|-
> > > $\Delta$ ATE|0.02|0.00|0.08
> > > $\Delta$ RPE_t|0.01|0.00|0.05
> > > $\Delta$ RPE_R [deg]|-0.18|-0.20|-0.28
> > >
> > > Nevertheless, we followed your suggestion and designed an even stronger experiment to evaluate this. Using the CARLA simulator, we generated six ego-motion trajectories across different maps (Towns 01, 02, 03, 04, 05, and 10HD). For each trajectory, we created two simulations: one with dynamic traffic participants and one with only static content. We rendered RGB, depth, poses, and dynamic masks at 10 FPS. The table below shows again that the pose accuracy with dynamic objects is comparable to the static-only setting (showing the difference in the metrics, as in the table above):
> > >
> > > $\Delta$ Metric between static and dynamic | <= t+0.6s | <= t+1.0s | <= t+2.0s | Avg
> > > -|-|-|-|-
> > > $\Delta$ ATE| -0.040 | -0.037 | -0.064 | -0.047
> > > $\Delta$ RPE_t| 0.005 | 0.003 | -0.004 | 0.001
> > > $\Delta$ RPE_R [deg]| -0.075 | -0.078 | -0.064 | -0.072
> > >
> > > ## Beyond-driving generalization
> > > Since our depth and pose transformers were trained only on driving data, fine-tuning indoors was necessary as the domain gap was otherwise too large. The indoor experiments provided in the rebuttal show that FR3D is not specific to driving settings but is versatile to work across diverse domains and motion patterns.
> > >
> > > ## Uncertainty/multi-modality
> > > We agree that uncertainty estimation and multi-modality are important for safety-critical deployment. However, these are substantial research challenges that represent distinct contributions from ours (requiring entire papers to address them). In our paper, we introduce this method as a first step in this future 3D reconstruction problem. We will further clarify this roadmap in the final version.

---

### Official Review · Reviewer_QSfy · 2026-03-13

**Soundness:** 3
**Presentation:** 4
**Significance:** 3
**Originality:** 3
**Overall Recommendation:** 5
**Confidence:** 4

**Summary:**

This paper proposes a novel 3D world model named FR3D, designed for future dynamic 3D reconstruction based on monocular observations. The authors propose to disentangle the camera ego-motion and environmental dynamics, and predict each of them separately. The model shows promising results in the experiment.

**Compliance With Llm Reviewing Policy:**

Affirmed.

**Final Justification:**

The rebuttal fully addressed my concerns. The new experiment supports the 'disentangle between camera trajectory and scenes' argument well, and I really like this design. Considering the overall technical solidness and experiments, I am happy to make an accept recommendation.

**Key Questions For Authors:**

Please refer to weaknesses.

**Limitations:**

The limiations are well-covered.

**Strengths And Weaknesses:**

Strength:
1) Disentangled Ego-Motion and World Motion: The model introduces two separate masked transformers (Pose Masked Transformer and Spatial Masked Transformer) that share information to predict camera poses and scene geometry independently. This decoupling ensures high physical and geometric consistency, even for future horizons up to 2 seconds.
2) Autoregressive Robustness: The paper designs an autoregressive training mechanism with a sliding window, requiring the model to process noisy, self-predicted future scene tokens during training. This strategy enhances the model's tolerance to input noise and effectively mitigates error drift during long-sequence inference.

weaknesses:
1) Although the paper shows promising results on camera prediction, but it lacks experiment to show the environment dynamics is really disentangled from camera motions. For example, can the author build inference cases with fixed camera token? In this way, the predicted scene should be evolving in a fixed viewpoint, which means the static part should be exactly the same. Without this experiment, one can only tell the disentangling of camera from environment, but not environment from camera.
2) The consistency between the predicted camera and the predicted scene is not shown. The authors should show the camera poses predicted by the model is consistent to the camera estimated from the predicted depth maps.

Dispite the above weaknesses, I still believe this paper has a great contribution. Thus, I tend to positively recommend this paper. However, due to the existing weaknesses, I cannot strongly advocate for it at current time.

---

> ### Author Rebuttal · Authors · 2026-03-31
>
> Thank you for your encouraging review. We appreciate that you recognize our paper as having a “great contribution”. Please find below our reply to your comments, where we grouped those targeting the same aspect.
>
> ## 1 Disentanglement experiment suggestion
> Thank you for this great idea. We took on your suggested experiment to show the disentanglement of the environment from the camera. Given a fixed camera, we verify that the environment dynamics are disentangled from the camera motion by measuring relative depth change for static and dynamic regions, respectively. First, we filtered the Waymo Open Dataset yielding a subset of scenes without ego-camera motion. Given a source depth frame $D_{source}$ and a target depth frame $D_{target}$, we measure relative depth change via the mean absolute log-ratio and convert it to percentage. The resulting values are reported below for static and dynamic regions, with and without frame-wise scale alignment, where $D_{source} = D_{t+0.2s}$ and $D_{target} \in \{D_{t+0.4s}, D_{t+0.6s}, D_{t+1.0s}\}$. Static and dynamic regions are separated by using the motion mask $M = M_{source} \cup M_{target}$, where $M_{source}$ and $M_{target}$ are the GT motion masks of the source and target frame, respectively.
>
> Region | Frame-wise Scale Aligned | t+0.4s | t+0.6s | t+1.0s
> -|-|-|-|-
> static | ✘ | 1.32 | 1.87 | 6.73
> static | ✔ | 0.40 | 0.634 | 1.64
> dynamic | ✘ | 8.41 | 12.55 | 17.82
> dynamic | ✔ | 7.64 | 11.43 | 12.97
>
> As shown in the scale consistency experiment to Reviewer oa97 (see rebuttal Point 11), we observe a slight drift when forecasting at longer horizons. This can be seen with static regions changing over time and with the relative change between source and target slightly increasing over time. By aligning the frame-wise scale the drift in static regions between source and target frame becomes 0.4 - 1.6%. Intuitively, frame-wise alignment reduces the relative depth change for dynamic regions less than for static regions.
> In summary, regardless of whether the scale alignment is applied or not, static regions in FR3D's depth prediction generally expose much less relative change (in %) than dynamic regions.
>
> ## 2 Predicted camera-scene consistency and dynamic shape stability
> Following your suggestion, we conducted a second experiment, where we consider all scene types irrespective of the existing ego-motion. Given $D_{source} = t + k$ and $D_{target} = t+k+1$, we transform $D_{source}$ to $D_{target}$ employing the FR3D’s predicted camera transformation and compute the relative depth change for static and dynamic regions as before. So, we always evaluated the relative depth change from $t+k$ to $t+k+1$. Static and dynamic regions are identified following the same approach as in the first experiment.
>
> Region | t+0.2s -> t+0.4s | t+0.4s -> t+0.6s | t+0.8s -> t+1.0s
> -|-|-|-
> static | 2.71 | 2.41 | 2.45
> dynamic | 5.94 | 5.26 | 4.40
>
> Additionally, we performed an experiment to assess the shape stability of dynamic objects for 4D consistency evaluation. First, we extract the predicted 3D point cloud of each dynamic object at time $t$ and $t+1$ using 4D consistent instance labels from the Waymo val split. Both are transformed into the same coordinate frame by compensating the ego-motion given predicted ego poses. Moreover, we remove object motion, by estimating the rigid transformation between both object point clouds using ICP. Then, we remove monocular scale drift by estimating a single global scale correction from static background pixels. Finally, we measure the remaining difference between both aligned point clouds using symmetric Chamfer distance. We compare the shape stability of CUT3R’s and FR3D’s dynamic reconstruction.
>
> Method | t+0.4s | t+0.6s | t+1.0s
> -|-|-|-
> CUT3R | 0.148 | 0.143 | 0.146
> FR3D | 0.157 | 0.141 | 0.141
>
> Both methods show very similar performance, with nearly identical Chamfer distances across all time steps. The error remains stable as the temporal gap increases, indicating consistent shape preservation over time.

---

> > ### Author Rebuttal · Reviewer_QSfy · 2026-04-02
> >
> > I appreciate the authors' effort in addressing my concerns. I will raise my score to 5.

---

### Official Review · Reviewer_p9ox · 2026-03-21

**Soundness:** 3
**Presentation:** 3
**Significance:** 3
**Originality:** 3
**Overall Recommendation:** 4
**Confidence:** 4

**Summary:**

The submission introduces a 3D world model that forecasts future depth and camera poses from monocular video, by predicting in the latent space of a frozen feed-forward 3D reconstructor (like CUT3R). It separates ego-motion and scene dynamics, using two masked transformers—one for pose tokens and one for spatial tokens, and use cross-attention to couple them, then decodes with the teacher’s heads to obtain multi-view-consistent depth and poses. Training uses latent teacher–student distillation with a self-rolling autoregressive sliding window to reduce the drift. Trained on Waymo and evaluated zero-shot on KITTI and nuScenes, FR3D surpasse DINO-Foresight and a CUT3R-foresight baseline up to 2 seconds ahead, and offer stable long-horizon geometry.

**Compliance With Llm Reviewing Policy:**

Affirmed.

**Final Justification:**

Thanks the authors for the detailed responses. My concerns are addressed with experiments, i.e., 4D consistency. I tend to accept this submission.

**Key Questions For Authors:**

1. How would the method incorporate explicit uncertainty or multimodal future hypotheses (for example, the distributional outputs, diffusion/flow in latent space), and how would these affect the training and evaluation?

2. To reduce dependence on a single reconstruction teacher, have you explored multi-teacher distillation or partial joint finetuning, and how does teacher model's quality or domain shift impact ceilings and failure modes?

3. Can the framework be extended with object-level dynamics (for example, scene flow, tracking tokens, or deformable fields) to better handle lateral motion and crowded scenes, and what supervision or priors may be required?

**Limitations:**

yes

**Strengths And Weaknesses:**

Strengths

1. Clear 3D inductive bias:

The method operates in a persistent 3D latent space, maintaining geometric consistency over long horizons.

2. Ego and world motion disentanglement:

The dual masked transformers (pose vs. spatial) with cross-attention could reduce self-motion and world-motion ambiguity.

3. Efficient training:

The latent teacher–student distillation reuses a frozen 3D reconstructor’s encoder, decoder and heads, lowering data and compute needs.

4. Robust rollouts:

The autoregressive sliding-window training mitigates drift, and increases robustness to noisy teacher tokens.

5. Comprehensive evaluation:

The evaluation of this submission includes ablations, static vs. dynamic region analysis, and qualitative long-horizon cases.

Weaknesses

1. Dynamic object modeling:

The method may struggle with lateral motions, which may show bias toward longitudinal motion.

2. Teacher dependency:

The performance ceiling may be tied to the chosen 3D reconstruction teacher (e.g., CUT3R). And the domain gaps in teacher can limit gains.

3. No explicit uncertainty:

The deterministic regression lacks multimodal futures and calibrated confidence.

4. Limited action conditioning:

The ego-motion are treated as latent proxy rather than conditioning on planned controls or goals.

5. Metrics scope:

The metrics mainly focuses on depth/pose. There are fewer evaluations of full 4D consistency, occlusion handling, or object-level trajectories.

6. Scalability to richer dynamics:

There are no explicit object-level motion fields or scene flow, which may hinder complex, crowded scenes.

---

> ### Author Rebuttal · Authors · 2026-03-31
>
> Thank you for your thoughtful review, recognizing FR3D having a “clear 3D inductive bias” and identifying our disentanglement “could reduce self-motion and ego-motion ambiguity”. We are encouraged you found our method “efficient”, “robust”, and “comprehensive[ly] evaluat[ed]”.  Please find below our reply to your comments, grouped by topic.
>
> ## 1 Bias to longitudinal motion
> Indeed, as discussed in Section 5 under “Limitations” and shown in Figure 5, in the rare cases when the ego car remains static, FR3D struggles to model full lateral motion of other dynamic agents. Additionally, we qualitatively verified that, unlike our model (trained only on Waymo), CUT3R (trained on 32 datasets with diverse motions) does not struggle with lateral movements. Given that and our teacher-student distillation, our model should have the potential to handle these cases well; yet, lateral motion scenarios’ low frequency relative to the dominant longitudinal motion patterns leads to a bias there. We clarify this in the final version and plan a more detailed investigation (e.g., via data rebalancing) as future work, but will include in the final version the qualitatives from CUT3R with lateral motion.
>
> ## 2 Extensions to improve lateral motion and crowded scenes
> We also thought about this idea and found it really interesting. FR3D could be theoretically extended to forecast scene flow tokens with a dedicated masked transformer or predict spatial tokens that already include this information (e.g., using Any4D [1] as the 3D reconstruction teacher, thereby learning its latent space instead of CUT3R’s). This could indeed help with challenging motion patterns, but cannot verify it now due to the limited time.
>
> ## 3 Dependency on a single teacher
> Thank you for the interesting suggestion. We did not explore multi-teacher distillation because FR3D learns to operate on the latent space of a single teacher model so we can leverage its generalizable downstream task heads frozen; so distilling info of multiple teachers would mix/overlay their latent/feature spaces. We believe this would deteriorate FR3D’s performance when reusing the frozen downstream task heads.
>
> ## 4 Teacher impact under domain shifts and failures
> We agree that biases or particular errors of the teacher may propagate to FR3D. However, our goal is to bring into the future the up-to-present predictions of a frozen, feed-forward 3D reconstruction teacher model (e.g., CUT3R), estimating how the scene evolves. We learn to operate on the frozen model’s latent space to forecast “future scene tokens” and reuse its frozen camera and depth prediction heads. This naturally upper-bounds performance. Poor performance of the teacher may affect FR3D’s performance but not FR3D’s training as CUT3R was trained on the same data (i.e., Waymo).
> In fact, FR3D can mitigate CUT3R’s pose jittering (see Figure 1), thanks to its autoregressive strategy, which helps handling token noise.
>
> ## 5 Uncertainty and multi-modal future hypotheses
> We appreciate your suggestion and agree that exploring multi-modal future hypotheses is valuable. Ours is a first step in this direction: we introduce the task of future 3D dynamic scene prediction and reconstruction, propose a method to address it, and show under zero-shot settings and long-time forecasting horizons how this works. We consider such extensions as future work. To extend FR3D to multi-modal future hypotheses, one could implement a stochastic forecasting model in the 3D reconstruction teacher’s latent space., e.g., by diffusing its predicted 3D scene tokens $s$ using DiTs instead of masked transformers. The DiT would be conditioned on the continuous noise level $t$, the past scene tokens $s_{1:N}$, and the noisy future latents $s_{N+1}^{(t)}$. For training, the regression objective could be adapted to a conditional flow matching objective. During inference, we could sample $s_{N+1}^{(0)} \sim \mathcal{N}(0,I)$. We plan this as future work.
>
> ## 6 Ego-motion and action conditioning
> We agree that our model does not have a direct action conditioning mechanism at inference time. Instead, it uses the ego-motion as a proxy for action, as also highlighted by Reviewer oa97. Essentially, the next timestamp’s ego-camera motion determines where the agent should move in the scene, and by predicting this ego movement separately from the motions within the scene (due to dynamic objects/agents), our model understands its motion from the world motion. We will clarify the scope in the camera-ready version.
>
> ## 7 4D consistency
> Thank you for your suggestion. We added a quantitative evaluation on 4D consistency in the rebuttal to Reviewer QSfy, Point 2.
>
> ## References:
> [1] Karhade et al., “Any4D: Unified Feed-Forward Metric 4D Reconstruction”, arXiv 2025.

---

> > ### Author Rebuttal · Reviewer_p9ox · 2026-04-03
> >
> > Thanks the authors for the detailed rebuttal. My concerns are addressed with experiments, i.e., 4D consistency. I tend to accept this submission.

---

### Official Review · Reviewer_VXw9 · 2026-03-22

**Soundness:** 2
**Presentation:** 3
**Significance:** 3
**Originality:** 2
**Overall Recommendation:** 4
**Confidence:** 2

**Summary:**

The paper introduces FR3D, a dynamic 3D world model designed to address physical inconsistencies in 2D video generation. By explicitly decoupling ego-motion from scene dynamics within a 3D latent space, the authors propose predicting persistent 3D implicit representations for future scene reconstruction.

**Compliance With Llm Reviewing Policy:**

Affirmed.

**Final Justification:**

I am raising my score to a Weak Accept.

The paper proposes a highly original and significant approach to 3D world modeling by explicitly disentangling ego-motion to improve spatiotemporal consistency. Initially, I questioned the soundness of the experiments due to missing baselines, unknown computational overhead, and potential long-term drift.

The authors' rebuttal directly resolved these issues. They clearly explained why 3DGS methods are not direct competitors for this specific task and provided new comparisons against DINO-Foresight and CUT3R-Prompt. They also shared concrete efficiency metrics (5.75GB VRAM and 0.85s latency) which proved the model is practically deployable. Finally, the added shape stability experiments cleared up my concerns regarding autoregressive error accumulation.

The rebuttal was strong, addressed my main critiques, and convinced me to change my evaluation. Overall, the method is solid and highly relevant for autonomous agents.

**Key Questions For Authors:**

1. How does FR3D mechanistically suppress state drift and error accumulation during long-term autoregressive 3D latent prediction? Please provide ablation results on shape stability over extended time horizons.
2. What is the actual computational cost for deployment? Please provide a detailed profiling of VRAM, FLOPs, and a latency comparison against pure 2D predictive models.
3. Can the authors provide quantitative comparisons with current SOTA 3D-aware dynamic forecasting methods, such as 4D representations or dynamic 3DGS-based predictors?

**Limitations:**

The authors fail to candidly discuss the substantial memory and computational bottlenecks inherent in 3D implicit state prediction. Furthermore, the paper omits failure cases in complex scenarios (e.g., dense multi-agent occlusions), leaving the true boundaries of the model's capability unexplored.

**Strengths And Weaknesses:**

Strengths
1. Solid Motivation: Effectively addresses the spatiotemporal consistency issues prevalent in 2D-based world models by leveraging 3D latent priors.
2. High Potential: The focus on resolving deformation and physical inconsistency holds significant value for long-term planning in autonomous driving and embodied AI.

Weaknesses
1. Modeling Flaws in Dynamic Agents: The autoregressive prediction of multiple high-frequency, non-rigid dynamic agents in 3D latent space is prone to error accumulation. The paper lacks rigorous ablation studies on shape preservation for long-sequence predictions.
2. Weak Baselines: The experimental comparison relies on outdated baselines (e.g., Copy Last, CUT3R-Foresight). It fails to include recent SOTA 3D-aware video prediction models or dynamic 3D Gaussian Splatting (3DGS) approaches.
3. Efficiency Black Box: There is a lack of transparency regarding Peak VRAM, FLOPs, and inference latency. For a model targeting "autonomous agents," the computational overhead of maintaining high-resolution 3D latent grids is a critical omission.

---

> ### Author Rebuttal · Authors · 2026-03-31
>
> Thank you for your feedback and appreciating our paper’s “solid motivation” and “high potential”. Please find below our reply to your comments, where we grouped those targeting the same aspect.
>
> ## 1 Limited baselines comparisons
> We appreciate your suggestion to strengthen the baselines. However, we believe there was a misunderstanding around what our method does. Here we explain the task we address and why there is no existing 3DGS-based predictor or 3D-aware dynamic forecasting method for this. The proposed method predicts future 3D scene tokens that it decodes into a coherent 3D point cloud and ego camera pose; it works for dynamic scenes and it disentangles ego motion from scene motion. These characteristics set it apart from existing works, represented in the following table:
>
> Category|Example|Predicts video|Predicts pose|Predicts depth|Can handle dynamics|Predict Gaussians|Predicts future
> -|-|-|-|-|-|-|-
> 2D world model|Vista [1]|✔|✘|✘|✘|✘|✔
> Feed-forward Gaussian reconstruction|AnySplat [2]|✘|✔|✔|✘|✔|✘
> Feed-forward dynamic 3D reconstruction|Depth Anything 3 [3]|✘|✔|✔|✔|✔|✘
> Feed-forward 4D Reconstruction|Any4D [4]|✘|✔|✔|✔ (scene flow)|✘|✘
> 2.5D forecasting| DINO-Foresight|✘|✘|✔|✔|✘|✔
> Online 3D reasoning|CUT3R|✘|✔|✔|✔|✘|✘
> Dynamic 3D world model for reconstruction|FR3D|✘|✔|✔|✔|✘|✔
>
> Given that, our baselines are up-to-date and follow the common practice in the forecasting domain (e.g., with Copy Last) and CUT3R-Foresight is an extension of DINO-Foresight [NeurIPS 2025], so it is from Dec. 2025, with FR3D submitted in Jan. 2026.
>
> Nevertheless, to strengthen our evaluation, we add here three baselines: CUT3R-Prompt, DINO-Foresight as reference, and Liu et al. [5]. CUT3R-Prompt utilizes CUT3R‘s state readout mechanism to estimate the point map based on GT poses provided as input. Liu et al. as additional comparison on KITTI. Please note that CUT3R-Prompt relies on GT poses and that its state readout mechanism was trained on 32 datasets while ours inherits CUT3R’s generalization capability only training on Waymo highlighting our training strategy’s effectiveness. Moreover, Liu et al. was trained and evaluated on KITTI while we evaluate zero-shot on it. As shown below, our FR3D outperforms all of these strong baselines and is competitive to Liu et al.
>
> Waymo - Depth (AbsR / δ1) | t+1.0s | t+2.0s
> -|-|-
> CUT3R-Prompt | 0.198 / 0.690 | 0.228 / 0.620
> DINO-Foresight | 0.155 / 0.800 | 0.232 / 0.678
> FR3D | **0.131** / **0.840** | **0.152** / **0.800**
>
> KITTI - Depth (AbsR / δ1) | Source Dataset | t+0.2s | t+0.6s
> -|-|-|-
> Liu et al. | KITTI | **0.099** / 0.895 | **0.108** / **0.878**
> DINO-Foresight (Zero-Shot) | Waymo | 0.010 / 0.904 | 0.128 / 0.857
> FR3D (Zero-Shot) | Waymo | **0.099** / **0.898** | 0.116 / 0.868
>
> ## 2 Runtime and efficiency
> Thank you for your suggestion. We conducted a profiling of peak VRAM and inference latency comparison of FR3D against Vista [1], a recent 2D video world model for driving scenes. We report inference latency when forecasting 2 seconds into the future and 4 context frames.
>
> ||VRAM (GB)|Latency (s)
> -|-|-
> Vista | 11.60 | 24.57
> FR3D | 5.75 | 0.85
>
> We further include a detailed profiling of the latency of FR3D's individual key steps.
> Module | Latency (ms)
> -|-
> Context Step (CUT3R) | 0.044
> Forecast Step (FR3D) |  0.038
> Head Prediction Step (CUT3R or FR3D) |  0.017
> Context for 4 frames (CUT3R) | 0.175
> Forecast for 10 frames (FR3D) | 0.545
> Total (FR3D) | 0.854
>
> We used a NVIDIA A100-SXM4-40GB GPU for all profiling experiments.
>
> ## 3 State drift suppression and shape preservation
> Thank you for your comment. We conducted a 4D consistency evaluation including 3D consistency analysis for static regions and shape stability for dynamic objects in the rebuttal of Reviewer QSfy, Point 2.
>
> ## References:
> [1] Gao et al., “Vista: A Generalizable Driving World Model with High Fidelity and Versatile Controllability”, NeurIPS 2024.
>
> [2] Jiang et al., “AnySplat: Feed-forward 3D Gaussian Splatting from Unconstrained Views”, SIGGRAPH Asia 2025.
>
> [3] Lin et al., “Depth Anything 3: Recovering the Visual Space from Any Views”, ICLR 2026.
>
> [4] Karhade et al., “Any4D: Unified Feed-Forward Metric 4D Reconstruction”, arXiv preprint arXiv:2512.10935 (2025).
>
> [5] Liu et al., “Meta-Auxiliary Learning for Future Depth Prediction in Videos”, WACV 2023.

---

> > ### Author Rebuttal · Reviewer_VXw9 · 2026-04-02
> >
> > The authors addressed my main concerns. The latency and VRAM metrics clear up the deployment questions, and adding the DINO-Foresight and CUT3R-Prompt baselines strengthens the empirical section. I will raise my score.

---

### Decision · Program_Chairs · 2026-04-30

**Decision:**

Accept (regular)

**Comment:**

This paper presents FR3D, a dynamic 3D world model that disentangles ego-motion and world dynamics in the latent space of a pretrained 3D foundation model (CUT3R). It effectively tackles geometric and temporal inconsistencies in world modeling and demonstrates a principled, scalable approach to dynamic 3D reconstruction. The rebuttal provided by the authors was strong, adding new baselines (e.g., DINO-Foresight, CUT3R-Prompt), efficiency profiling, and quantitative validations of shape stability and 4D consistency. Most reviewers upgraded their ratings and endorsed acceptance after the revisions. One reviewer gave a weak reject, asking for additional analyses. The authors provided the corresponding evidence after rebuttal. AC finds the additional evidence is sufficient, especially on the disentanglement issue. Nonetheless, all reviewers acknowledge that FR3D is technically sound, original, and impactful, advancing dynamic 3D world modeling and its applications in embodied AI. AC recommends acceptance after considering all the reviews and additional materials.